# Defining essential charged residues in fibril formation of a lysosomal derived N-terminal α-synuclein truncation

Ryan P. McGlinchey[1], Sashary Ramos[1], Emilios K. Dimitriadis[2], C. Blake Wilson [3] & Jennifer C. Lee [1]✉

N- and C-terminal α-synuclein (α-syn) truncations are prevalent in Parkinson's disease. Effects of the N- and C-terminal residues on α-syn aggregation and fibril propagation are distinct, where the N-terminus dictates fibril structure. Here, the majority of α-syn truncations are assigned by intact mass spectrometry to lysosomal activity. To delineate essential charged residues in fibril formation, we selected an N-terminal truncation (66–140) that is generated solely from soluble α-syn by asparagine endopeptidase. Ala-substitutions at K80 and E83 impact aggregation kinetics, revealing their vital roles in defining fibril polymorphism. K80, E83, and K97 are identified to be critical for fibril elongation. Based on solid-state NMR, mutational and Raman studies, and molecular dynamics simulations, a E83–K97 salt bridge is proposed. Finally, participation of C-terminal Lys residues in the full-length α-syn fibril assembly process is also shown, highlighting that individual residues can be targeted for therapeutic intervention.

Abundance of α-synuclein (α-syn) fibrils is a pathological hallmark of Parkinson's disease (PD), multiple system atrophy (MSA) and dementia with Lewy bodies[1,2]. Recent biochemical and structural studies on α-syn fibrils from patient-derived materials have revealed distinct fibril conformers, suggesting a link between fibril structure and disease phenotype[3–7]. Importantly, seed amplification assays are potential diagnostic tools in synucleinopathies[8–10]. Although limited, current data point to different cellular environments and/or biomolecules (e.g. phospholipids[11] and polyphosphate[12]) as playing a role in dictating fibril structure[13,14].

While α-syn is a 140-amino-acid protein, truncated species have been identified in disease[15], where regions of either the C- or N-terminus are missing. Specifically, N-terminally truncated (ΔN) α-syn variants, 5–140, 39–140, 65–140, 66–140, 68–140, and 71–140, and C-terminally truncated (ΔC) α-syn variants, 1–101, 1–103, 1–115, 1–119, 1–122, 1–133 and 1–135, have been found in brains of PD patients[16–18]. Based on our own data[19–21], we suggest the majority of these truncations originate in the lysosome from proteolytic activities by cathepsins (Cts) B, L, D and asparagine endopeptidase (AEP). From a structural perspective, we have shown C-terminal truncations 1–103 and 1–122 result in modest conformational changes[22], while the N-terminal truncation 41–140 adopts a completely different fibril core[23] when compared to full-length protein[22]. This infers that the unresolved N-terminal residues from recombinant fibrils play a role since their absence changes the structure as shown in 41–140[23]. Furthermore, structures obtained from patient-derived fibrils contain additional N-terminal residues, supporting this region in playing a significant role in dictating fibril structure[3,4,6].

In vitro studies using recombinant α-syn have aided in understanding the aggregation process[23–31]. Both intra- and inter-molecular electrostatic interactions between the N-terminal (residues 1–60) and non-amyloid β component (NAC, residues 61–95) regions as well as between N- and C-terminal (residues 96–140) regions are widely recognized as early events in α-syn aggregation[32–36]. Structurally,

[1]Laboratory of Protein Conformation and Dynamics, Biochemistry and Biophysics Center, National Heart, Lung, and Blood Institute, National Institutes of Health, Bethesda, MD, USA. [2]Biomedical Engineering and Physical Science Shared Resource Program, National Institute of Biomedical Imaging and Bioengineering, National Institutes of Health, Bethesda, MD, USA. [3]Laboratory of Chemical Physics, National Institute of Diabetes and Digestive and Kidney Diseases, National Institutes of Health, Bethesda, MD, USA. ✉e-mail: leej4@nhlbi.nih.gov

nearly all α-syn fibrillar cores (~residues 36 to 99) solved from recombinant proteins contain at least one salt bridge[22,23,37–41]. This is perhaps not surprising given that there are a total of 12 charged residues (7 Lys, 4 Glu and 1 Asp) within this defined core region out of the 39 total charges in the full-length sequence (Fig. 1a). Moreover, the abundance of possible interactions could explain why full-length α-syn fibrils are so polymorphic. In supporting this notion, our own work

revealed that as more of the N-terminus is removed (up to residue 40 which removes 12 charged residues), the fibril population becomes more homogeneous[23]. However, molecular mechanisms that give rise to fibril polymorphism in α-syn variants remain to be elucidated at the residue level.

Given the abovementioned fact that a more homogenous fibril structure is produced upon the removal of N-terminal residues, we

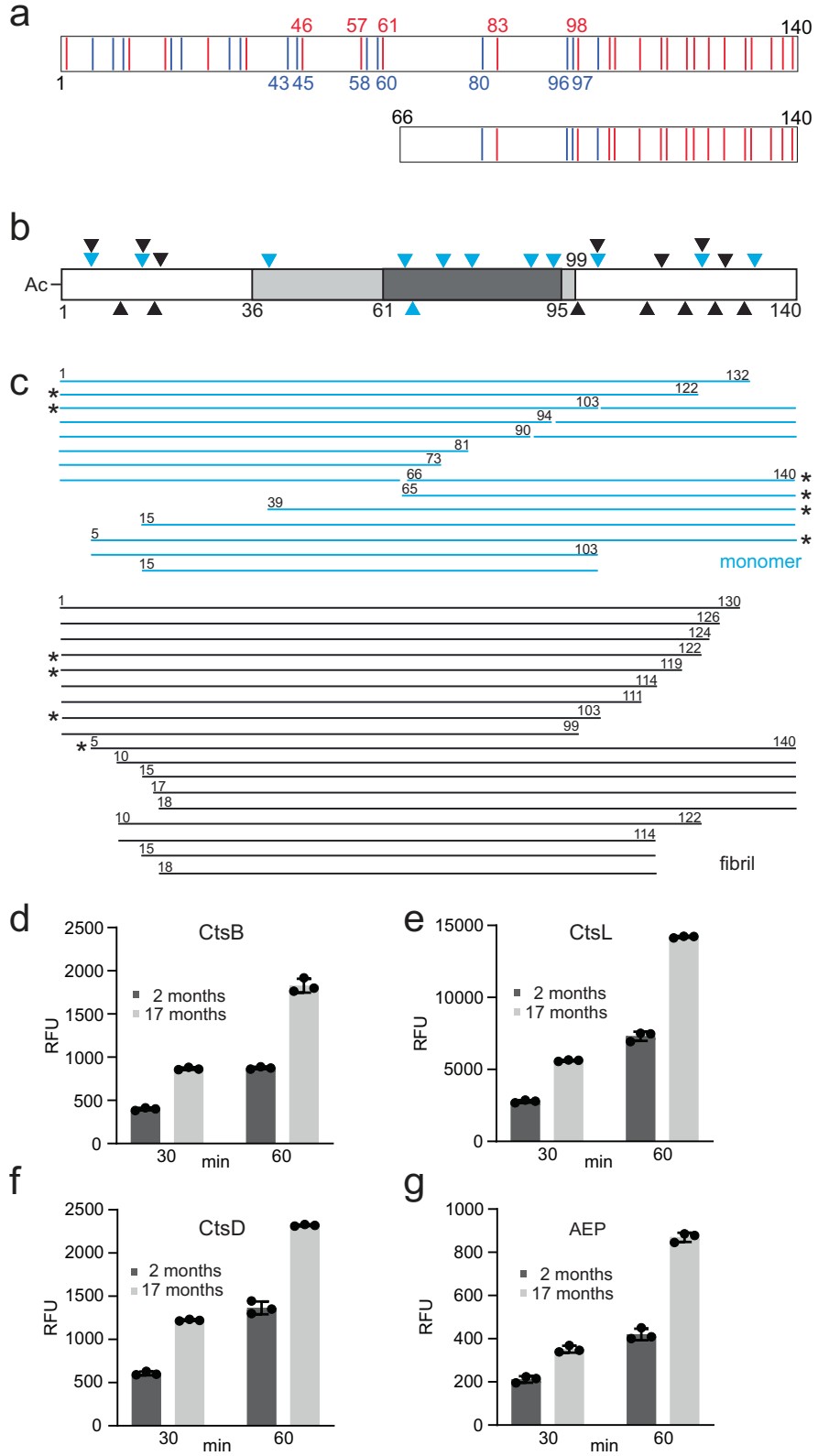

**Fig. 1 | Lysosomal degradation of α-syn. a** Schematic representation of the primary amino acid sequences of 1–140 (top) and 66–140 (bottom), coloring basic (blue) and acidic (red) residues. Charged residues that participate in salt bridges within the shared α-syn fibril core (residues 36–99) are indicated[3,4,6,7,11,22,23,37–41,44–46,55–57,61,79]. **b** Schematic representation of the primary amino acid sequence of α-syn with cleavage sites generated for either soluble (cyan) or fibrillar (black) α-syn. Residues 36–99 (light gray) show cryoEM fibril core while residues 61–95 denote the NAC region (dark gray). **c** α-Syn peptide fragments derived from lysosomal degradation of soluble (cyan) and fibrillar (black) α-syn. Brain lysosomal extracts were obtained from a 10-month-old male mouse. Fragment masses and residue assignments are reported in

Supplementary Table 1. Previously identified fragments from PD patients are denoted by asterisks. Effect of mouse age on specific lysosomal protease activities. Brain lysosomal extracts were obtained from two female mice aged 2 and 17 months. Fluorogenic substrates **d** Ac-RR-AMC for CtsB, **e** Ac-FR-AMC for CtsL, **f** MCA-GKPI-LEFRKL(Dnp)-D-R-NH₂ for CtsD, and **g** AENK-AMC for AEP were incubated with lysosomal extracts (10–40 µg total protein) at pH 5.0 with 5 mM DTT, 37 °C. Fluorescence was recorded as a function of time (30 and 60 min), and relative fluorescence units (RFU) are reported ($n = 3$ technical replicates). Data are presented as mean values ± SD. Source data are provided as a Source Data file.

rationalized that a ΔN-α-syn variant would be an appropriate model to study the role of specific charged residues in α-syn fibril polymorphism as clear morphological changes can be easily discerned. We chose a biologically relevant N-terminally truncated variant, 66–140[18] (Fig. 1a), by first establishing its origin from the degradation of the soluble state in the lysosome, a cellular site directly linked to PD[42]. Its pathological relevance was also ascertained by identifying AEP, which is upregulated in PD[43] as the sole lysosomal protease for generating 66–140. In addition, while this construct greatly reduces the number of possible charges, it retains the prevalent charged residue K80 that participates in salt bridges in multiple fibril structures[4,8,22,23,38,39,44–46] as well as the NACore (residues 68–78), a segment that is critical for aggregation and cytotoxicity[47].

A total of eleven 66–140 mutants were evaluated including six single-alanine (K80A, E83A, K96A, K97A, D98A, and K102A), two double-alanine (K96A/K97A and E104A/E105A) as well as three charge-switch (K80E/E83K, E83K/K96E, and E83K/K97E) mutants. Aggregation propensity, fibril propagation, and structure were characterized by thioflavin-T (ThT) fluorescence assays, Raman and solid-state NMR spectroscopies, transmission electron (TEM) and atomic force (AFM) microscopies. Our data reveal site-specific charged residues (K80, E83, K96, and K97) modulate the aggregation process of 66–140. By neutralizing E83, rapid fibril formation was observed, resulting in a complete morphological change from twisted to straight fibrils, invoking its vital role in dictating fibril structure. Residues K80 and K97 are identified to be essential for fibril elongation as determined by cross-seeding experiments. Importantly, we discovered that either K96 or K97 is necessary for amyloid formation as the double K-to-A mutant completely inhibits aggregation. Based on the fibril templating compatibility of charge-switch mutants with the wild-type fibril, a E83–K97 salt bridge is proposed, which was also observed in a model by molecular dynamics simulations. To provide evidence that the three key lysines, K80, K96, and K97, also participate in the aggregation of the full-length protein, three corresponding acetylated (Ac) full-length mutants AcK80A, AcK96A, and AcK97A were examined. Collectively, this work highlights the importance of multiple C-terminal Lys residues and their electrostatic interactions in the aggregation mechanism of α-syn.

## Results

### 66–140 originates from lysosomal degradation of soluble α-synuclein

To establish the importance and relevance for using 66–140 as a model for interrogating the effect of charged residues on fibril polymorphism, we first investigated its biological origin. Prior work[19–21] had suggested that the lysosome could be responsible for generating ~60% of the previously identified pathological truncations[16–18]. However, the assignments were based on identifiable cleavage sites generated from digestion of full-length α-syn using individual lysosomal proteases[19–21]. Here, we sought to observe these pathological truncations directly from lysosomal activity using purified mouse brain lysosomes and to address whether they are derived from a soluble or fibrillar state of α-syn. Liquid chromatography and mass spectrometry (LC-MS) was used

to identify peptide fragments generated by α-syn degradation reactions over time (Supplementary Table 1). Both cleavage sites and peptide fragments were compared between soluble and fibrillar α-syn.

Using soluble α-syn as the substrate, cleavage sites (denoted by X/X, where / indicates the cut) at F4/M5, G14/V15, L38/Y39, T64/N65, N65/V66, G73/V74, T81/V82, A90/A91, F94/V95, N103/E104, N122/E123 and G132/Y133 were identified (Fig. 1b). In the fibrillar state, N- and C-terminal cleavage sites were observed at F4/M5, S9/K10, G14/V15, V16/A17, A17/A18, Q99/L100, N103/E104, G111/I112, E114/D115, D119/P120, N122/E123, A124/Y125, E126/M127 and E130/E131 (Fig. 1b). α-Syn fragments which have been identified from patient material (5–140, Ac1–103 and Ac1–122, where Ac abbreviates for N-terminally acetylated) were common to both soluble and fibrillar α-syn (Fig. 1c). This supports the validity of using lysosomes in generating PD-related truncations. Notably, three previously identified N-terminal truncations 39–140, 66–140, and 68–140 from PD patients[14] are derived solely from digestion of soluble α-syn (Fig. 1c). These data establish that lysosomes harbor the proteolytic machinery to generate the N-terminal truncation 66–140 from the soluble state.

Next, we asked which protease(s) are responsible for generating 66–140 by using purified lysosomal proteases CtsB, CtsL, CtsD and AEP (Supplementary Figs. 1 and 2 and Supplementary Tables 2 and 3). Prior work suggested that AEP is likely involved, since it specifically recognizes and cleaves proteins containing asparagine[19]. As anticipated, complete peptide mapping using individual proteases revealed only AEP can selectively cleave soluble and not fibrillar α-syn at N65/V66 to generate 66–140 (Supplememtary Figs. 1 and 2). The presence of 66–140 is bolstered by literature where AEP levels are upregulated in PD models[43].

Finally, we assessed the impact of mouse age on lysosomal protease activity. The rationale is to ask whether truncation generation by lysosomes would be affected as α-syn levels are known to increase with age. Brain lysosomes were isolated from mice euthanized at 2 *vs.* 17 months of age, and fluorogenic substrates specific for CtsB (Fig. 1d), CtsL (Fig. 1e), CtsD (Fig. 1f), and AEP (Fig. 1g) were used to monitor individual protease activities. Results showed elevation of all four protease activities from 17 months old brain lysosomes, indicating that the potential for truncations is enhanced with age. It is important to note that both lysosomal proteases AEP[43] and variants of CtsB[48,49] are implicated in PD etiology. Collectively, our data indicate that the AEP derived N-terminal 66–140 truncation is pathologically relevant and warrants detailed investigation into its aggregation propensity.

### Fibril formation of 66–140

Aggregation kinetics of 66–140 were monitored by ThT fluorescence and compared to full-length Ac1–140 at two protein concentrations (40 and 80 µM, Fig. 2a). Typical sigmoidal curves were observed, consisting of lag, growth, and stationary phases, where increases in ThT intensity indicate amyloid formation[50]. Lag and growth phases are defined as the time period where there is minimal ThT intensity ($t_{lag}$) and the time to reach half-maximum ThT intensity ($t_{1/2}$), respectively. At both concentrations, $t_{lag}$ and $t_{1/2}$ of 66–140 are reduced compared to that of Ac1–140, indicating that removal of the first 65 residues resulted

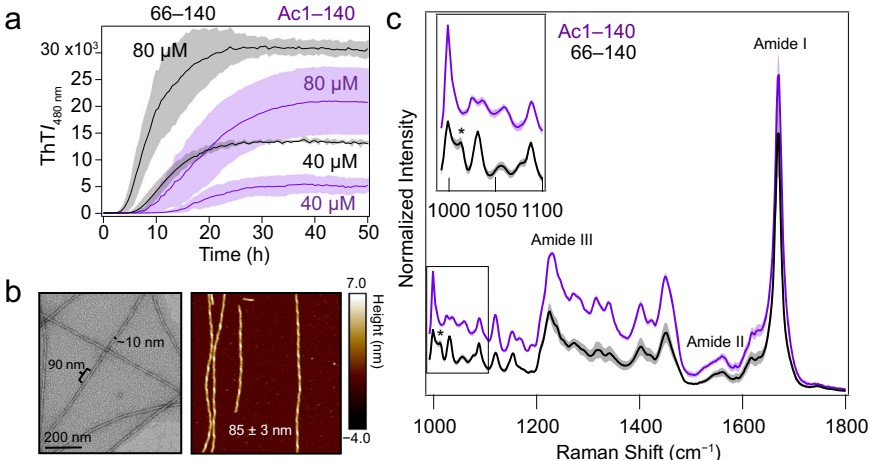

**Fig. 2 | Aggregation kinetics and structural characterization of 66–140 fibrils.**
**a** Aggregation reactions of Ac1–140 (purple) and 66–140 (black) monitored by ThT fluorescence in pH 7.4 buffer (20 mM NaPi, 140 mM NaCl) at 37 °C with continuous linear shaking supplemented with a 2-mm glass bead. Solid lines and shaded regions represent mean and SD, respectively ($n \geq 4$). For 66–140, average $t_{lag}$ and $t_{1/2}$ values were 5 and 12 h for 40 μM and 3 and 9 h for 80 μM, respectively. For Ac1–140, average $t_{lag}$ and $t_{1/2}$ values were 8 and 13 h for 40 μM and 5 and 14.5 h for 80 μM, respectively. **b** Representative TEM and AFM images of 66–140 taken post-aggregation. Scale bars as indicated. Full-sized images are shown in Supplementary Fig. 5. **c** Raman spectra of Ac1–140 (purple) and 66–140 (black) fibrils. Solid lines and shaded regions represent the mean and SD, respectively ($n \geq 15$). Amide-I, amide-II, and amide-III regions are as indicated. The frequency location of a distinguishing peak (1012.5 cm⁻¹) in 66–140 is denoted by an asterisk. Inset shows an expanded view of the boxed spectral area. Spectra have been normalized to the Phe breathing mode (1003 cm⁻¹). Difference spectrum is shown in Supplementary Fig. 6. Source data are provided as a Source Data file.

in faster aggregation under these experimental conditions. To establish the role of electrostatics in 66–140 aggregation, the solution ionic strength was varied, resulting in faster aggregation; both lag and growth phases shortened with increasing NaCl concentrations (Supplementary Fig. 3). Corresponding TEM images confirmed that fibrils were formed (Supplementary Fig. 3). To confirm a nucleation-polymerization mechanism, self-seeding experiments were performed for 66–140. Experimental conditions were modified such that protein alone did not aggregate by lowering protein concentration (30 μM) and reducing agitation by the omission of the glass bead (Supplementary Fig. 4a). As expected, the addition of pre-formed seeds (5 mol%) eliminated the lag phase, resulting in rapid growth. Faithful propagation of twisted fibril morphology was verified by TEM (Supplementary Fig. 4b).

## Structure characterization of 66–140 fibrils

Negative-stain TEM and AFM images taken post-aggregation (Fig. 2b) show mainly twisted fibrils with approximate widths of 10 nm and heights of 7 nm. TEM images show a predominantly twisted fibril with a helical pitch of ~90 nm, corroborated by similar values (85 ± 3 nm) determined by AFM analysis (Fig. 2b, Supplementary Fig. 5b), considerably more twisted compared to the full length protein, which has a helical pitch of 120 nm[22]. Structural differences were characterized in more detail by Raman spectroscopy, a vibrational spectroscopic technique shown to be sensitive to α-syn fibril polymorphs[51]. While indistinguishable amide-I peaks (1669 cm⁻¹) characteristic of β-sheet amyloid structure[52] were observed for both constructs, 66–140 exhibits sharpened amide-III (1226 cm⁻¹) and narrower amide-II (1560 cm⁻¹) bands, indicating an overall more ordered fibril structure compared to that of Ac1–140 (Fig. 2c). Additionally, spectral differences are observed in the fingerprint region (990–1200 cm⁻¹); in particular, the appearance of a new peak at 1012.5 cm⁻¹ is noted for 66–140. Difference spectrum is shown in Supplementary Fig. 6.

Definitive residue-level structural information was obtained by solid-state NMR experiments using ¹³C/¹⁵N-labeled 66–140 fibrils. Well-resolved peaks, with ¹³C linewidths of ~1 ppm, were recorded in 2D ¹³C–¹³C correlation spectra (Fig. 3a, Supplementary Fig. 7, and Supplementary Table 4). 2D NCACX and NCOCX as well as 3D NCACX,

NCOCX, and CaNCOCX (Supplementary Table 5, Supplementary Figs. 8–12) spectra were acquired. Assignments (Supplementary Table 5) were made for residues 66 to 95 utilizing a computational Monte Carlo/simulated annealing (MCSA) algorithm[53]. Secondary chemical shifts (Fig. 3b) are consistent with a predominantly β-sheet-like secondary structure. Based on TALOS-N[54] dihedral angle predictions, residues 69–72, 74–79, and 87–92 are classified as β-sheet, whereas the intervening residues (83–86) are more ambiguous, possibly indicative of a turn (Fig. 3b). Although residues K96 and K97 could not be unambiguously assigned in 66–140, it is clear that the observed NMR signals correspond to more than one lysine. The remaining C-terminal residues (98 to 140) could not be assigned, suggesting structural disorder in this region. Albeit slightly larger, a similar proteinase-K (PK)-resistant core of residues 66–113 was obtained by LC-MS (Supplementary Table 6). Of note, the ¹³C_δ carbon of E83 is within a salt-bridging distance (3.2–3.5 Å) to a ¹⁵N-atom of a lysine sidechain as determined by frequency-selective rotational-echo double resonance (FS-REDOR) experiments (Supplementary Fig. 13); however, as noted earlier, the specific lysine could not be identified.

Interestingly, there are quite good overlapping sections of assigned residues of 66–140 with the ssNMR structure of 1–140 amplified from human Lewy body dementia (LBD) tissue[7]. Specifically, residues T75-A78 and A85-T92 in 66–140 have similar ¹³Cα and backbone ¹⁵N chemical shifts that form the G73-F94 β-hairpin in the amplified LBD full-length structure[7]. Thus, we decided to use the NMR structures determined by Dhavale et al. (PDB ID: 8FPT[7]) as a starting point for generating a 66–140 protomer model. Model eight, of the ten determined, was selected as the best candidate due to the highest degree of overlapping secondary structure assignments; this model was trimmed to only include residues 66–103, and the secondary structure was adjusted to represent the β-sheet structure determined by our ssNMR experiments (Fig. 3c). As the ssNMR data suggest, a β-hairpin model with a turn formed by residues 79–87 is illustrated with flexible portions around residue 73 and following residue 92. Intriguingly, this β-hairpin conformation is similar to that of the Lewy fold from cryoEM structures of fibrils extracted from brains of patients with PD, PD dementia and LBD[4] as well as the extended hairpin conformation found in 41–140 composed of residues G68 to V95[23].

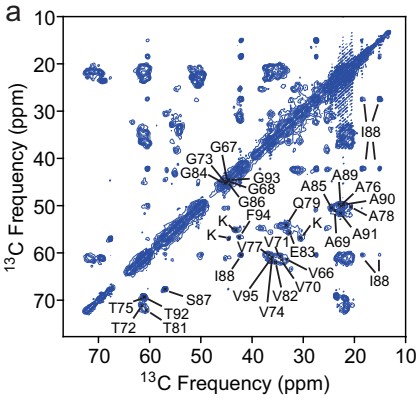
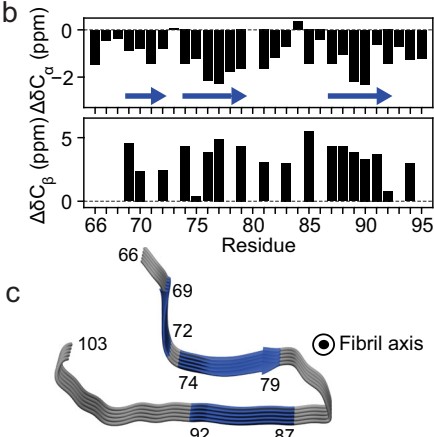

**Fig. 3 | ssNMR of 66–140 fibrils. a** 2D $^{13}$C-$^{13}$C ssNMR spectra of 66–140 fibrils, recorded under conditions where only immobilized, structurally ordered segments are visible. Residue assignments are based on 2D NCACX/NCOCX, and 3D NCACX/NCOCX/CANCOCX spectra (Supplementary Figs. 7–12). Contour levels increase by successive factors of 1.3. **b** Secondary chemical shifts $\Delta\delta^{13}$C$\alpha$ (top) and $\Delta\delta^{13}$C$\beta$ (bottom) obtained by subtracting random-coil shifts from the observed chemical shifts. Blue arrows indicate β-sheet secondary structure predicted by TALOS-N. **c** Structural model of the 66–140 fibril generated by modifying model 8 in PDB ID: 8FPT[7] using UCSF Chimera[77]. NMR-assigned residues consist of 66 to 95, with blue arrows depicting the β-sheet secondary structure for residues 69 to 72, 74 to 79, and 87 to 92. Fibril axis as indicated. Source data are provided as a Source Data file.

## Site-specific effects of charge neutralization on aggregation kinetics

The role individual charged residues play in fibril nucleation and growth was assessed by generating single-alanine mutants that neutralized charges within the fibrillar core (K80A and E83A), as well as outside the core (K96A, K97A, D98A, and K102A, Fig. 4a). Their aggregation propensities were compared to 66–140 (40 and 80 µM) (Fig. 4b). Values for $t_{lag}$ and $t_{1/2}$ are reported in Supplementary Table 7. Mutations at core positions, K80 and E83, had significant and varying effects on kinetics, whereas sites, D98 and K102, in the disordered region had little to no effect (Fig. 4b). For E83A, aggregation was hastened with markedly lower ThT signals at both protein concentrations. Contrastingly, Ala-substitutions at K80 and K97 protracted aggregation kinetics, impacting both lag and growth phases. Even after 160 h, K80A still does not reach stationary phase, but exhibited higher ThT intensity than 66–140 at both protein concentrations (Supplementary Fig. 14). For K96A, it has a slightly delayed aggregation at the lower protein concentration. While D98A recapitulated 66–140 kinetics, it exhibited higher ThT responses at both protein concentrations.

Quantification post-aggregation (160 h) showed predominately insoluble material, 76–83% and 80–89%, for all mutants at 40 and 80 µM, respectively (Supplementary Fig. 14). One exception was K97A which generated only 29% and 53% precipitates at 40 and 80 µM, respectively; while reduced ThT signals were observed, the fluorescence intensity differences were larger than would be expected based on pelleted protein amounts. Similarly, ThT signals for K80A, E83A, and D98A also do not correspond to relative fibril amounts, suggesting ThT sensitivity to fibril structural differences[27].

## Removal of charges at K80 and E83 alter fibril morphologies

Both TEM and AFM images taken post-aggregation revealed distinct fibril morphologies for K80A and E83A (Fig. 4c). Although still twisted like 66–140, K80A fibrils appear very heterogenous with varied helical pitches (Fig. 4c, Supplementary Fig. 15). Measured half-pitch lengths by TEM and AFM suggest several fibril polymorphs with a main helical pitch of 73 ± 5 nm (Supplementary Fig. 15). In contrast, E83A adopt mainly straight fibrils with a width of ~10 nm and height of ~6 nm, though a few twisted fibrils were observed by AFM (Supplementary Fig. 15). All other mutants (K96A, K97A, D98A, and K102A) looked more reminiscent to the 66–140 fibrils. Despite morphological

differences exhibited by K80A and E83A, all single-Ala mutants contained the 66–140 PK-resistant core, corresponding to residues 66–113 (Supplementary Table 8).

## Raman spectroscopy reveals distinctive features of K80A and E83A fibrils

Next, we turned to Raman spectroscopy to characterize the single-Ala mutants and compare them to 66–140 (Supplementary Fig. 16). In Fig. 5a, the amide-I, amide-III, and fingerprint regions are shown. As assessed by the amide-I band, all proteins exhibit β-sheet content (Fig. 5a, right panel), with K80A displaying a shift in the band as highlighted by the difference spectrum (Fig. 5b, right panel, blue trace). In the amide-III region, spectra for all mutants were nearly indistinguishable from 66–140 with the exception of E83A and K80A. Unlike the others, E83A has a smooth amide-III band with no discernable spectral features (Fig. 5a, middle panel). Upon inspection of the difference spectra for the amide-III band (Fig. 5b, middle panel), both E83A and K80A exhibit shifts to higher wavenumbers. Difference spectra of the fingerprint region clearly show spectral changes for K80A and E83A compared to 66–140 (Fig. 5b, left panel). For E83A, striking features include a red-shifted peak at 1054 cm$^{-1}$ along with disappearances of the shoulder at 1077.8 cm$^{-1}$ and the peak at 1012.5 cm$^{-1}$ (Fig. 5a, left panel). Interestingly, an altered band shape at 1012.5 cm$^{-1}$ was also observed for K80A; it is weaker and broadened compared to that of 66–140. While there is no definitive assignment of this vibrational frequency at 1012.5 cm$^{-1}$, it is clear that this peak is characteristic of the 66–140 fibril structure that is recapitulated by all mutants except K80A and E83A, which adopt different conformations. Taken together with TEM and AFM, our data identify K80 and E83 as key residues in establishing the amyloid structure of 66–140.

## E83A fibrils cannot cross-seed 66–140

To provide definitive answers as to whether fibril structures are changed by single charge neutralization, we evaluated whether these single-Ala mutant fibrils (1 mol%) could cross-propagate or cross-seed soluble 66–140. Because if a mutant fibril is able to template 66–140 fibril formation through the elimination of the lag phase, then it would show that it has highly similar and compatible fibril structure, and based on the abovementioned ultrastructural and Raman spectral data, we anticipate that all but K80A and E83A fibrils would be

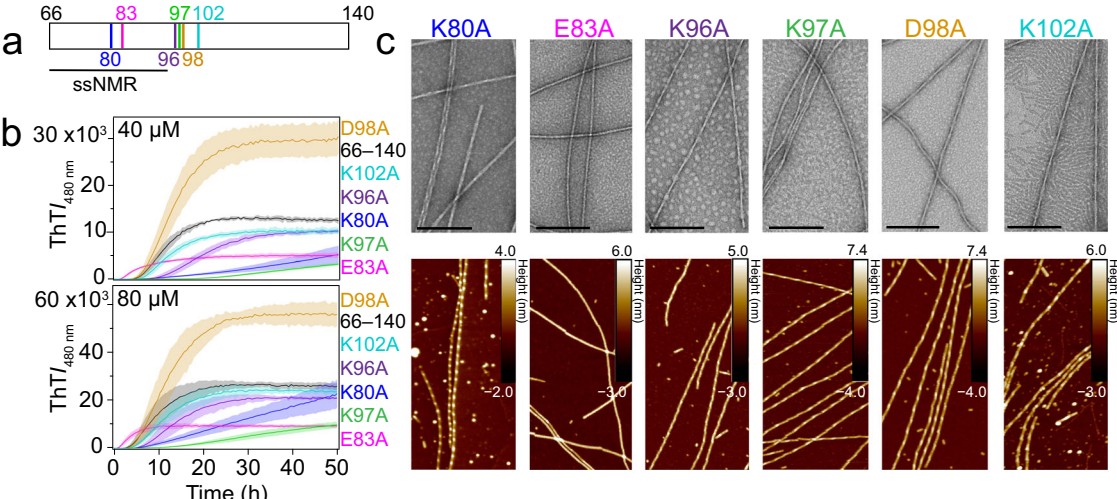

**Fig. 4 | Aggregation kinetics and structural characterization of single-Ala mutants of 66–140. a** Schematic representation of the primary amino acid sequences of 66–140 showing sites of mutations in this study. Fibril core assigned by ssNMR is underlined. **b** Aggregation reactions monitored by ThT fluorescence of 40 (*top*) and 80 (*bottom*) μM of 66–140 (black), K80A (blue), E83A (magenta), K96A (purple), K97A (green), D98A (gold) and K102A (teal) in pH 7.4 buffer (20 mM NaPi, 140 mM NaCl) at 37 °C with continuous linear shaking supplemented with a 2-mm glass bead. Solid lines and shaded regions represent mean and SD, respectively ($n \geq 5$). Complete aggregation kinetics to 160 h are shown in Supplementary Fig. 14. **c** Fibril morphologies of single-Ala mutants visualized by TEM (*top*) and AFM (*bottom*). TEM scale bars are 200 nm. Full-size images are shown in Supplementary Fig. 15. Source data are provided as a Source Data file.

competent seeds. Under these experimental conditions (30 μM and no beads), unseeded 66–140 aggregation does not occur and requires the presence of pre-formed fibrils to promote fibril propagation (Fig. 5c). As anticipated, the addition of E83A fibrils had no effect on the aggregation of 66–140 (Fig. 5c), even increasing seed concentration five-fold did not result in any changes (Supplementary Fig. 17). However, K80A fibrils unexpectedly exhibited similar cross-seeding abilities along with other single-Ala fibrils with all reactions proceeding to near completion (95–99%), though there were moderate differences in growth rates.

### K80, E83, and K97 are key residues for fibril propagation

To investigate directly whether and which specific charged residues impact fibril propagation of 66–140, the reversed cross-seeding experiments were performed, where pre-formed 66–140 fibrils (1 mol%) were added to soluble single-Ala mutants (Fig. 6a). Upon cross-seeding, the two most C-terminal mutants, D98A and K102A, had nearly identical kinetic profiles and aggregation yields (73–75%) compared to that of the self-seeded 66–140 (73%), indicating that these sidechains in the disordered region contribute minimally to the fibril formation process, in accord to abovementioned results. However, since the PK-resistance core (residues 66 to 113) is larger than that of the ssNMR (residues 66 to 95) core, we ascertained that the double E104A/E105A mutant could be easily seeded by 66–140 fibrils, verifying that these residues are not involved in fibril propagation (Supplementary Fig. 18).

By moving closer into the structured core, K96A and K97A both have measurable effects. While K96A exhibited comparable kinetics, it yielded a lower amount of fibrils formed (47%), whereas 66–140 fibril propagation of K97A struggled with markedly slowed kinetics and minimal fibril yield (13%). In light of these results, we hypothesized that the two neighboring Lys residues have some compensatory effect for each other. Indeed, a double K96A/K97A mutant proved that in the absence of either Lys residue, fibril propagation was not possible with seeding (Supplementary Fig. 19a). Moreover, the aggregation propensity of K96A/K97A is greatly suppressed, again highlighting the essential role of electrostatic interactions in facilitating aggregation of 66–140 (Supplementary Fig. 19b). The fact that K96A/K97A completely

shuts down fibril formation suggests that under these experimental conditions, at least one of these two Lys residues are necessary to form amyloids.

Consistently, soluble K80A inhibited fibril formation, resulting in minimal yield (19%) despite displaying an unusually high ThT signal, which does not represent the total amount of aggregated material as documented earlier (Supplementary Fig. 14). Even at higher seed concentrations (5 mol%), kinetics of the two worse mutants, K80A and K97A, were unchanged (Supplementary Fig. 20). While cross-seeding K96A and K97A with 66–140 fibrils had low yields, TEM images revealed these fibrils are twisted and similar to that of the 66–140 fibrils (Fig. 6b and Supplementary Fig. 21). However, the twisted fibrils for cross-seeded K80A appeared eclectic with different fibrils having varying half-pitches, unlike 66–140 (Fig. 6b and Supplementary Fig. 21). Albeit with significantly lower efficiency (16% aggregated material), soluble E83A is able to template onto 66–140 fibrils, adopting twisted fibrils (Fig. 6b and Supplementary Fig. 21) unlike that of those de novo straight E83A fibrils (Fig. 4c). This is a surprising result when considering the reverse cross-seeding reaction shown in Fig. 5c, where the native soluble protein cannot be seeded by E83A fibrils.

To evaluate the fidelity of fibril structure templating, these cross-seeded fibrils were separated by ultracentrifugation and investigated by Raman spectroscopy. For comparison, the spectra of the unseeded and cross-seeded fibrils are overlaid and their difference spectra are shown (Fig. 6c and Supplementary Fig. 22). Although K96A and K97A yield less fibrillar material, the spectra remain highly similar to the wild-type, suggesting little change in the structure upon cross-seeding in accord with the TEM images. As shown previously, K80A and E83A have unusual Raman spectral features supporting distinctive structure polymorphs induced by these mutations. When cross-seeded with 66–140 fibrils, key changes in the spectra are indicated by dashed lines in Fig. 6c. Notably, the peak at 1012.5 cm⁻¹ is now clearly evident in both mutants. In addition, the cross-seeded E83A spectrum nearly resembles that of 66–140 fibrils with a shifted peak to lower energy (1056.5 → 1054 cm⁻¹) and the appearance of the peak at 1077.5 cm⁻¹, verifying that structural templating has succeeded. Likewise, the spectrum of cross-seeded K80A in the fingerprint region suggests it has adopted similar

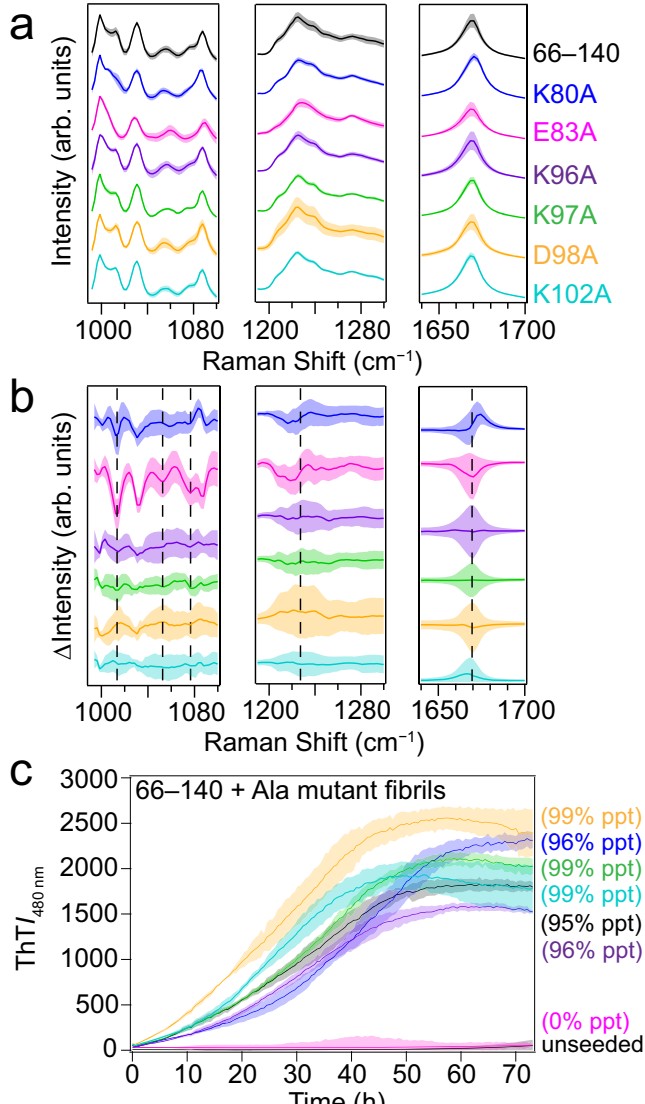

**Fig. 5 | Raman spectroscopy of single-Ala mutants and cross-seeding kinetics of 66–140 by single-Ala fibrils. a** Raman spectra of 66–140 (black) and mutants (K80A (blue), E83A (magenta), K96A (purple), K97A (green), D98A (gold) and K102A (teal)) in the fingerprint (*left*), amide-III (*middle*) and amide-I (*right*) regions. Solid lines and shaded regions represent the mean and SD, respectively ($n \geq 15$). Spectrum of 66–140 is the same as shown in Fig. 2c. Full spectra are shown in Supplementary Fig. 16. **b** Difference spectra generated by subtracting the average spectrum of 66–140 from that of the 66–140 mutants. Shaded regions represent the propagated error. Dashed lines are guides to key spectral differences in the mutants. Color scheme is same as Fig. 5a. **c** Aggregation kinetics monitored by ThT fluorescence (10 μM) of soluble 66–140 (30 μM) seeded with 0.3 μM 66–140 (black), K80A (blue), E83A (magenta), K96A (purple), K97A (green), D98A (gold) and K102A (teal) fibrils in pH 7.4 buffer (20 mM NaPi, 140 mM NaCl) at 37 °C with continuous linear shaking in the absence of 2 mm glass beads. Unseeded 66–140 is shown as a control. Solid lines and shaded regions represent mean and SD, respectively ($n \geq 6$). *ppt* refers to amount of insoluble material pelleted by ultracentrifugation. Source data are provided as a Source Data file.

fibril structures as of the 66–140 fibrils, although the TEM images reveal ultrastructural heterogeneity.

## A proposed E83–K97 salt bridge in 66–140 fibril structure

While we have clearly shown that charged residues modulate aggregation kinetics and fibril propagation efficiency of 66–140, the question remains as to which Lys (80, 96, or 97) residue is interacting with

E83 as indicated by the ssNMR data. To address this question, we tested the cross-seeding compatibility of three K-to-E charge-switch mutants (K80E/E83K, E83K/K96E, and E83K/K97E) with 66–140 fibrils (Fig. 7a). The underlying premise is that even if reversed, the correct salt bridge pair would faithfully template on to 66–140 fibrils and propagate fibril growth. Pre-formed 66–140 seeds (1 mol%) were added to the three charge-switch mutants (30 μM). Of the mutants tested, only E83K/K97E are seeded by 66–140 fibrils, bypassing the lag phase and exhibiting rapid growth kinetics as the 66–140 self-seeded reaction (Fig. 7a). Fidelity of fibril templating were confirmed at both the ultrastructural and molecular level by TEM and Raman spectroscopy, respectively (Supplementary Figs. 23 and 24). Figure 7b shows twisted fibrils with helical pitches between 85–90 nm, similar to 66–140 fibrils. Finally, identical overlapping Raman spectra were observed for the cross-seeded and the unseeded 66–140 sample (Fig. 7c). Based on these results, we suggest that K97 is the most likely lysine involved in a salt bridge with E83.

## Observation of E83–K97 interactions in a 66–140 model structure by molecular dynamics simulations

To gain further clarity on the lysine residue participating in a salt bridge with E83, molecular dynamics simulations were conducted on the model presented in Fig. 3c. For the simulations, a solvated, two-protofibril model with the same interface as that reported by Dhavale et al. was used[7]. The equilibrated starting structure and the final structure obtained after 30 ns are shown in Supplementary Fig. 25a; interestingly, the final structure shows the development of an overall twist that is not observed in the starting structure. The last 15 ns of the trajectory, where the root mean square deviation has stabilized, were analyzed to determine interactions between charged residues. Radial pair distribution functions ($g(r)$) were calculated to determine the density of K80, K96, and K97 residues within a distance of 0 to 10 Å from E83 (Supplementary Fig. 25b); these results show only K97 has a significant density between at 3.2–4 Å, supporting a possible salt bridge interaction between E83–K97 (Supplementary Fig. 25c). The trajectory was further analyzed to identify charged residues within a salt-bridging distance (3.2–4 Å) of each other. The most common residues involved in salt bridges were D98, K96, K97, and E83, with D98 and K96 forming intramolecular salt bridges and K97 and E83 interacting between the two protofibrils. The observed E83–K97 interaction appears dynamic, with the salt bridge breaking and reforming throughout the trajectory and not all chains forming the salt bridge at the same rate, as illustrated in Supplementary Fig. 25d. This is not surprising as K97 is located on the more flexible part of the protofibril. Overall, the modeling results suggest only K97 interacts with residue E83 in a way that forms a salt bridge.

## Effect of charge neutralization of C-terminal lysines on Ac1–140 aggregation

To support 66–140 is an advantageous model that offers relevant insights in studying α-syn fibril formation, we turned our attention to the acetylated full-length protein. We asked whether single K-to-A mutations in the C-terminus (K80A, K96A, and K97A) will also modulate fibril formation of the full-length protein as observed in 66–140, in establishing their outright importance even in the presence of competing N-terminal Lys residues (Fig. 8a). Comparing to Ac1–140 (Fig. 8b), aggregation kinetics data (50 μM protein) clearly showed that neutralizing all three C-terminal Lys positions, K80 (Fig. 8c), K96 (Fig. 8d) and K97 (Fig. 8e), had significant effects. For AcK80A, rapid aggregation and a lower ThT response were observed, whereas both AcK96A and AcK97A exhibited noticeable well-to-well variation, but overall slower aggregation compared to that of Ac1–140. Even at 100 μM, both AcK96A and AcK97A displayed well-to-well variation with both high and low ThT responses, while K80A is still faster than Ac1–140 (Supplementary Fig. 26).

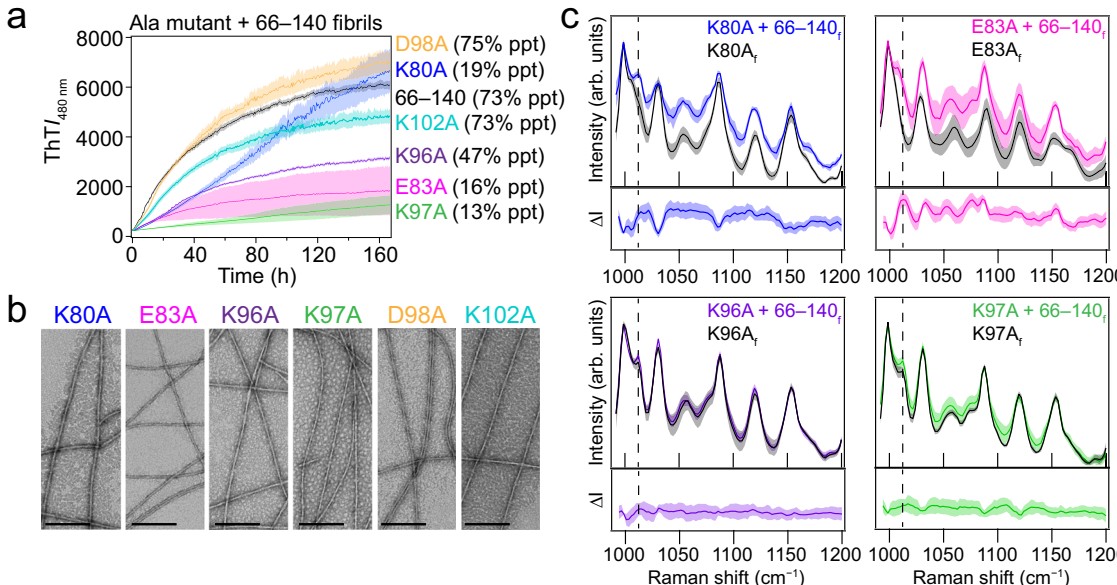

**Fig. 6 | Cross-seeding kinetics, TEM, and Raman spectroscopy of single-Ala mutants seeded with 66–140 fibrils. a** Aggregation kinetics monitored by ThT (10 μM) of soluble 66–140 (black), K80A (blue), E83A (magenta), K96A (purple), K97A (green), D98A (gold) and K102A (teal) (30 μM) seeded with 66–140 fibrils (0.3 μM) in pH 7.4 buffer (20 mM NaPi, 140 mM NaCl) at 37 °C with continuous linear shaking in the absence of 2 mm glass beads. Solid lines and shaded regions represent mean and SD, respectively (n ≥ 12). Precipitated (ppt) material calculated (%) after ultracentrifugation post-aggregation is shown for each mutant. **b** Representative TEM images taken post-seeding. Scale bar is 200 nm. Full images

are shown in Supplementary Fig. 21. **c** Comparison of Raman fingerprint region of the cross-seeded (as colored in Fig. 6a) and unseeded Ala-mutant fibrils (black). Solid lines and shaded regions represent the mean and SD, respectively (n ≥ 15). Dashed lines indicate the frequency location of a distinguishing peak (1012.5 cm⁻¹) in 66–140. Spectra of the unseeded samples are the same as shown in Fig. 5a. Full spectra are shown in Supplementary Fig. 22. The bottom panel shows difference spectra generated by subtracting the average unseeded from the average seeded spectrum. The error has been propagated from the SD of the averaged spectra. Subscript f denotes fibrils. Source data are provided as a Source Data file.

TEM images taken post-aggregation revealed distinct fibril morphological features for all three mutants (Fig. 8f and Supplementary Fig. 27). Compared to wild-type fibrils which appear mostly rod-like, the mutant fibrils were twisted with varying half-pitches (AcK80A: 80 and 100 nm, AcK96A: 100 and 150 nm, and AcK97A: 80 nm). These morphological differences are also corroborated by Raman spectroscopic differences in the fingerprint region (Supplementary Fig. 28a) for all three mutants, indicating conformational differences between the K-to-A mutants and the wild-type fibrils. Most noticeably, the amide-I peak of AcK97A is shifted to a lower frequency compared to the wild-type fibrils and the other mutants, and an additional feature at 1039 cm⁻¹ is also observed for this mutant (Supplementary Fig. 28b). Both AcK97A and AcK96A have shifts to higher frequency in the feature at 1056.4 cm⁻¹, indicating underlying conformational changes that are different to that of the wild-type and AcK80A fibrils (Supplementary Fig. S28c).

To determine seeding compatibility, cross-seeding experiments were performed (Fig. 8g). Experimental conditions were modified where beads were absent during aggregation that resulted in unseeded Ac1–140 aggregation being significantly reduced and requires the presence of pre-formed fibrils to promote fibril propagation. While AcK96A and AcK97A fibrils showed a seeding effect by eliminating the lag phase, a protracted growth phase was observed when compared to self-seeding by Ac1–140. In contrast, a lag phase remained when AcK80A fibrils were used, indicating that this mutant has a different structural conformation in which wild-type protein cannot propagate; however, there appears to be enhancement of aggregation *via* secondary nucleation.

## Discussion

At the residue level, the mechanisms that give rise to α-syn fibril polymorphism remain ill-defined. Elucidating the molecular origins of

how polymorphs are established is of high biological significance as it could reveal the amyloid structure-disease relationship. From the burgeoning fibril structures determined by cryoEM, one emerging hypothesis is that charged residue interactions that result in salt bridges could dictate the formation of different α-syn fibril polymorphs. Formation of a specific salt bridge could be caused by genetic (e.g. missense mutation[44–46,55–57] and alternative splicing[58–60]) and environmental (e.g. post-translational modification[61], pH[62], and ionic strength[63]) factors. To date, nearly all α-syn structures contain at least one salt bridge that results from the preponderance of Lys, Glu and Asp residues within the α-syn amyloid core (Fig. 1a).

Due to the polymorphic nature of full-length α-syn, we elected to study the effect of specific charged residues in α-syn fibril formation by using an N-terminal α-syn truncation 66–140, whose fibrillar state is highly homogeneous. While this truncation was previously identified in PD patients, herein we show that 66–140 originates from the proteolysis of soluble α-syn by lysosomal AEP, which is upregulated in PD[43]. This data along with other PD-related truncations (e.g. 5–140, 39–140, 65–140, 68–140, Ac1–103, and Ac1–122) being derived from the soluble state by cathepsin activity (Supplementary Fig. 1), reiterates the involvement of lysosomal function in generating other amyloidogenic α-syn variants and could bear pathological consequences. From a mechanistic point of view, we propose that an interplay exists between the different types of truncations, where levels of protease activity dictate which truncations are generated and to what extent. Thus, altering lysosomal activity such as AEP could offer a targeted strategy to modulate α-syn homeostasis. Furthermore, given that ΔN-truncated fibrils appear to be poor seeds for soluble full-length α-syn, whereas the removal of C-terminal residues has the opposing effect, where it creates potent seeds that can efficiently propagate soluble full-length α-syn[19,20], it is imperative that future studies consider contributions from both fibril variants.

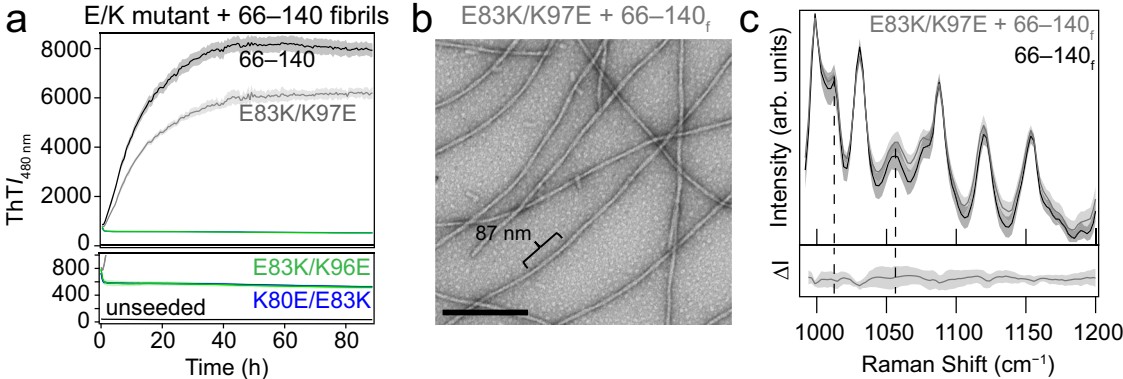

**Fig. 7 | Aggregation and structural characterization of charge-switch mutants.**
**a** Aggregation kinetics of 30 µM soluble 66–140 (black), K80E/E83K (blue), E83K/
K96E (green), and E83K/K97E (gray) seeded with 66–140 fibrils (1 mol%) in pH 7.4
buffer (20 mM NaPi, 140 mM NaCl) at 37 °C with continuous linear shaking in the
absence of 2 mm glass beads. Solid lines and shaded regions represent mean and
SD, respectively (*n* ≥ 6). **b** Representative TEM image taken post aggregation of
E83K/K97E seeded with 66–140 fibrils. Full image is shown in Supplementary
Fig. 23. **c** Comparison of Raman spectra of the fingerprint region of cross-seeded

E83K/K97E with 66–140 (gray) fibrils and 66–140 (black) fibrils. Solid lines and
shaded region represent the mean and SD, respectively (*n* ≥ 15). The bottom panel
shows the difference spectrum generated by subtracting the spectrum of 66–140
from that of cross-seeded E83K/K97E. Dashed lines are guides to key spectral fea-
tures in 66–140. Spectrum of 66–140 is the same as shown in Fig. 2c. Full spectrum
of cross-seeded sample is shown in Supplementary Fig. 24. Subscript f denotes
fibrils. Source data are provided as a Source Data file.

Through mutational studies of 66–140, we have identified four
important specific charged residues (E83, K80, K96, and K97) and
delineated their roles in the fibril propagation and structure. Neu-
tralizing K80, E83, K96 and K97 had obvious effects on the lag phase,
which is interpreted to signify the involvement of these sidechains in
primary nucleation, in which polypeptides undergo conformational
changes and oligomerize to generate a nucleus for fibril formation.
While charge neutralization of E83 facilitated nucleation, charge neu-
tralization of the Lys residues had the opposing, inhibitory effect.
While it was expected that Ala-substitutions at K80 and E83 would have
strong influences as they are found within the ssNMR structure core,
the dramatic effects of K96A and K97A were not as they reside outside
in the disordered region. These results demonstrate that residues
outside the β-sheet-containing core can be just as important as those
forming the core.

The importance of the three C-terminal K80, K96, and K97 was
also validated in the full-length protein, where both AcK96A and
AcK97A mutants mirror the sluggish behaviors of the corresponding
66–140 mutants. Interestingly, Ala-substitution of K80 had opposite
behaviors for full-length and 66–140, where aggregation kinetics is
accelerated and slowed, respectively. Nevertheless, the results support
66–140 as a useful model for investigating α-syn fibril formation,
offering relevant sites that modulate aggregation propensity.

Notably, the Ala-substitution of E83 in 66–140 is associated with a
remarkable morphological change to straight fibrils (Fig. 4c) that are
incompetent seeds for soluble 66–140 (Fig. 5c). This striking obser-
vation indicates that the carboxylate side chain of the native E83 in 66–
140 cannot template the E83A fibril structure. The loss of the dis-
tinctive Raman peak at 1012.5 cm⁻¹ for E83A (Fig. 5b), may suggest this
peak represents a E83 to Lys salt bridge in the 66–140 structure. While
the ssNMR data support this claim, to date, no Raman peak assignment
for a salt bridge has been made. Interestingly, the full-length protein
structure has been shown to contain two salt bridges, yet no
1012.5 cm⁻¹ peak is evident from the Raman spectrum. However, a
shoulder at 1008 cm⁻¹ is observed in AcK97A and AcK80A, highlighting
the conformational sensitivity of this spectral region. Collectively,
these data support that a salt bridge is present, involving E83, which
dictates the 66–140 structure because, in its absence, a different
polymorph is formed.

Once nucleation occurs, fibril elongation can proceed where
structural rearrangements ensue as soluble proteins add on to fibril

ends. Initial fibril structure is preserved because the fibril ends act as
conformational templates for the incoming proteins. Neutralizing
K80, E83, and K97 dramatically reduced the ability of 66–140 fibrils to
cross-seed, resulting in minimal fibrils formed (13–19%, Fig. 6a). Inef-
ficiency of fibril propagation of the native 66–140 structure by E83A
makes sense because E83A forms a distinct fibril polymorph which is
incompatible with 66–140. The low fibril yields of K80A and K97A
suggest that positive sidechains are indeed essential for fibril elonga-
tion, which is also reflected in the moderate yield obtained for K96A
(47%). These results affirm that side chains of K80 and K97 as well as
E83 are critical in the process of 66–140 fibril growth, which could
involve their respective electrostatic interactions. Importantly, resi-
dues K80 and K97 also appear to play a role in fibril propagation of the
full-length protein with Ala-substitution at K80 exhibiting a pro-
nounced reduction in efficiency (Fig. 8g). Clearly, more work would be
necessary to understand the exact interactions responsible for the
inhibitory effects.

In defining which Lys residue interacts with E83, we showed that
only E83K/K97E out of the three charge-switch mutants was able to
template onto the 66–140 fibrils (Fig. 7), supportive that E83–K97 salt
bridge is present and required for fibril propagation as the other
charge-switches are incompatible with 66–140 structure. While not
entirely validated by structural determination, molecular dynamic
simulations indicate only K97 and E83 can form a salt bridge between
two protofibrils (Supplementary Fig. 25). These data expand on the
number of already reported salt bridges from α-syn structures to
include K97 with E83, further exemplifying the polymorphic nature of
α-syn fibrils.

In conclusion, our work on a lysosomal-derived N-terminal α-syn
truncation shows that individual charged residues are drivers of
aggregation, and we have identified key residues for fibril propaga-
tion efficiency. Broadly, the effect of these mutants reiterates the
importance of electrostatic interactions, likely through salt bridge
formation, in establishing α-syn fibril polymorphism. We also
demonstrate that changes in fibril structure and faithful templating
of the fibril seed can be readily observed by Raman spectroscopy, a
simple and efficient method to characterize distinct molecular fea-
tures. Moving forward, future work will need to test the relevance of
electrostatic interactions on patient-derived fibrils once conditions
have been optimized to ensure correct propagation of the original
fibril polymorph as studies have shown that in vitro amplication can

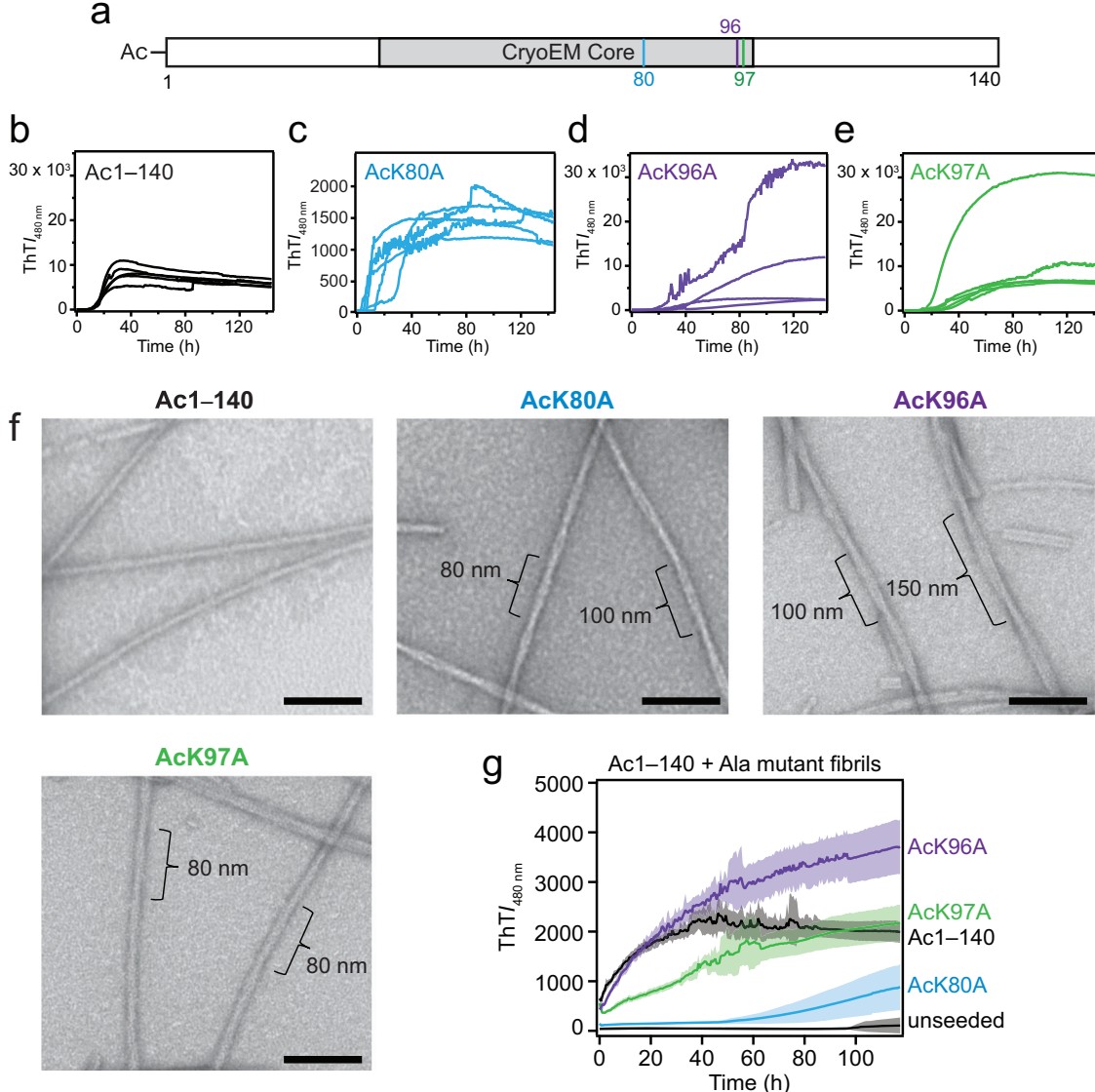

**Fig. 8 | Aggregation kinetics, fibril morphology characterization, and cross-seeding kinetics of single-Ala mutants of Ac1–140. a** Schematic representation of the primary amino acid sequences of Ac1–140 showing sites of mutations. CryoEM fibril core assigned to residues 36 to 99 (gray). Aggregation reactions monitored by ThT fluorescence (10 μM) of 50 μM Ac1–140 (**b**), AcK80A (**c**), AcK96A (**d**) and AcK97A (**e**) in pH 7.4 buffer (20 mM NaPi, 140 mM NaCl) at 37 °C with continuous linear shaking supplemented with a 2-mm glass bead ($n \geq 4$). **f** Fibril morphologies of single-Ala mutants visualized by TEM. Scale bars are 100 nm. Full-size images are shown in Supplementary Fig. 27. **g** Aggregation kinetics monitored by ThT fluorescence (10 μM) of soluble Ac1–140 (50 μM) seeded with 2.5 μM Ac1–140 (black), K80A (blue), K96A (purple) and K97A (green) fibrils in pH 7.4 buffer (20 mM NaPi, 140 mM NaCl) at 37 °C with continuous linear shaking in the absence of 2 mm glass beads. Unseeded 66–140 (black) is also shown as a control. Solid lines and shaded regions represent mean and SD, respectively ($n \geq 5$). Source data are provided as a Source Data file.

result in different structures[13,64]. Research efforts also should be extended to evaluate the role of electrostatic interactions in other amyloids[65]. Defining specific charged residues necessary for fibril propagation will offer viable targets for intervention with small molecules that could serve as a potent therapeutic strategy against these debilitating human diseases.

## Methods

### Ethical statement

Mice study was performed with approval from the National Human Genome Research Institute (NHGRI) Animal Care and Use Committee using NHGRI G-05-4 protocol.

### Reagents

Chemicals were obtained from Sigma unless otherwise noted.

### Protein expression and purification

Primers used is this study are provided in a Source Data file. N-terminally acetylated full-length proteins were expressed in *E. coli* BL21(DE3) (New England BioLabs) using human α-syn (pRK172)[66] and (pT7-7)[67] and yeast NatB genes[68] and purified by two rounds of anionic chromatography after heat treatment and acidification[69]. These constructs contain a silent mutation (TAT) at residue 136 to avoid the spontaneous mutation of Tyr-to-Cys[70].

Plasmids for α-syn 66–140 and Ala-mutants (K80A, E83A, K96A, K97A, D98A, K102A, E104A/E105A, K96A/K97A and E83K/K97E) were constructed such that they generated sequences with a N-terminal hexa-His-tag followed by an enterokinase (EK) recognition site that leaves no overhang after proteolytic cleavage. Plasmids for K80E/E83K and E83K/K96E mutants were purchased from Genscript, USA. PCR amplified fragments and synthesized genes were cloned into a

pET21a(+) vector. These constructs contain a silent mutation (TAT) at residue 136 to avoid the spontaneous mutation of Tyr-to-Cys[70]. Constructs were verified by Sanger sequencing (Psomagen, USA).

Proteins were expressed in *E.coli* BL21(DE3) (New England Bio-Labs). Shaken cultures (1-L Luria Broth) were grown at 37 °C to an $OD_{600}$ ~ 0.5–0.8 and then induced with 1 mM IPTG for 3–4 h. For $^{13}C^{15}N$-labeled α-syn, cells were grown in M9 media containing D-glucose (U-$^{13}C6$) and $^{15}NH_4Cl$ (Cambridge Isotopes Laboratories) according to previously published protocols[71]. Cells were collected by centrifugation (11,160 × *g*, Sorvall SLC-6000 rotor) for 20 min at 4 °C and then resuspended in denaturing buffer (6 M guanidine hydrochloride, 100 mM NaCl, 100 mM $Na_2HPO_4$, pH 7.5) and incubated overnight at 4 °C. Lysate was then spun at 104,444 × g (Beckman Ti45) for 45 min at 4 °C. The pellet was discarded, and the supernatant was applied to a HisPrep FF 16/10 column (GE Healthcare) using an ÄKTA Pure chromatography system (GE Healthcare). The column was first washed with 3-column volumes of buffer A (6 M urea, 100 mM NaCl, 100 mM $Na_2HPO_4$, pH 7.5 and 10 mM imidazole). Protein was eluted using a gradient method composed of buffer A and buffer B (6 M urea, 100 mM NaCl, 100 mM $Na_2HPO_4$, pH 7.5 and 500 mM imidazole). First, 1-column volume of 4% buffer B, followed by a linear gradient to 30% buffer B, and finally a step gradient to 100% buffer B. The eluted protein was desalted into 10 mM Tris-HCl, 50 mM NaCl, pH 7.4 buffer using a HiPrep desalting column (GE Healthcare).

For removal of the His-tag from 66–140 and mutants, a His-tagged Enterokinase (EK) protease (Genscript, USA) was used. To activate EK protease activity, 2 mM $CaCl_2$ was added. All His-tagged proteins were initially incubated with 50 µL EK (1 mg/mL) in 40–50 mL (10 mM Tris-HCl, 50 mM NaCl, pH 7.4 buffer) at RT with cleavage efficiency monitored by SDS-PAGE and LC-MS analysis (NHLBI Biochemistry Core). Additional protease was added if needed to completely remove the His-tag. After full cleavage, the protein sample was reapplied to a HisPrep FF 16/10 column and the flow through fractions collected. Final purification was achieved by strong anionic exchange chromatography (MonoQ HR 16/10 column, GE Healthcare) using a linear NaCl gradient in 20 mM Tris, pH 8. The protein eluted at ~230 mM NaCl.

Sample homogeneity and identity were evaluated using SDS-PAGE and LC-MS (NHLBI Biochemistry Core). Measured masses were: 14,502 Da for Ac1–140; 14,445 Da for AcK80A, AcK96A, and AcK97A; 7757 Da for 66–140, K80E/E83K, E83K/K96E, E83K/K97E; 7700 Da for K80A, K96A, K97A and K102A; 7699 Da for E83A; 7713 Da for D98A; 7643 Da for K96A/K97A and 7641 Da for E104A/E105A. Protein concentrations were determined using a molar extinction coefficient estimated on basis of amino-acid content: $\varepsilon_{280\,nm} = 4470\,M^{-1}\,cm^{-1}$ for 66–140 and mutants and $\varepsilon_{280\,nm} = 5960\,M^{-1}\,cm^{-1}$ for Ac1–140. All purified proteins were aliquoted and stored at −80 °C until use. All buffers were filtered (0.22 µm).

## Lysosome isolation from mice brains
C57BL/6 mice were housed and bred under NHGRI Animal Care and Use Committee–approved NHGRI G-05-4 protocol, which followed the NIH policy documents related to animal care and use [http://oacu.od.nih.gov/NIHpolicy/CC614.pdf]. Brains were first washed in ice-cold PBS followed by gently homogenizing using a Dounce homogenizer (50 strokes) on ice. Samples were spun at 500 *g* for 10 min at 4 °C, and soluble lysates were combined with OptiPrep (lysosomal enrichment kit, Pierce) to a final concentration of 15% and placed on top of a discontinuous density gradient (17, 20, 23, 27, and 30%). Samples were centrifuged for 2 h at 145,000 × *g* at 4 °C. The enriched lysosomal band on the top was collected, diluted 3-fold with PBS, and pelleted by centrifugation for 30 min at 16,100 × *g* at 4 °C. Lysosomes were washed once with PBS, resuspended in pH 5 buffer (50 mM sodium acetate, 20 mM NaCl), and centrifuged again at 16,100 × *g* for 30 min at 4 °C. Lysosomes were

stored at −80 °C until use. Before use, samples were repeatedly thawed and frozen for four times. Bradford assay (detergent compatible kit from Pierce Biotechnology) was used to determine protein concentration for the following lysosomal extracts: 2 month (0.3 mg/mL), 10 month (6 mg/mL) and 17 month (0.3 mg/mL).

## Cathepsin activities determined by fluorogenic substrates
Cathepsin activities in lysosomal extracts (1 µg total protein) were measured using fluorogenic substrates in pH 5 buffer supplemented with 5 mM DTT: 200 µM Ac-RR-AMC (catalog no. 219392, Calbiochem) for CtsB, 200 µM Ac-FR-AMC for CtsL (catalog no. 03–32-1501–25MG, Calbiochem), 20 µM MCA-GKPILEFRKL(Dnp)-D-R-NH2 (catalog no. 219360–1MG, Calbiochem) for CtsD, and 200 µM z-AAN-AMC for AEP (catalog no. 4033201, Bachem). Flourescence measurements (technical replicates of 3) were performed in polypropylene 384-well flat-bottom microplates (Greiner Bio-One) containing 50 µL sample at 37 °C using a microplate reader (Tecan Infinite M200 Pro) at 30 and 60 min. Excitation (360 nm) and emission (460 nm) wavelengths was used for CtsB, CtsL, and AEP, and excitation (328 nm) and emission (393 nm) wavelengths was used for CtsD.

## Degradation reactions of recombinant Ac1–140
In microcentrifuge tubes (1.5-mL Protein LoBind tubes, catalog no. 022431081, Eppendorf), soluble α-syn and preformed α-syn fibrils (15 µM) were incubated with lysosomal extracts. The amounts of lysosomal extracts were 30–120 µg for soluble and 120 µg for fibrillar α-syn digestion. Limited proteolysis of soluble and fibrillar α-syn (15 µM) was performed using either CtsB (Sigma-Aldrich, C8571-25UG), CtsL (Sigma-Aldrich, C6854-25UG), CtsD (Sigma-Aldrich, C8696-25UG) and AEP (R and D Systems catalog no. 2199-CY-010) and monitored over time. Final protease concentrations were 15 nM for soluble and 300 nM for fibrillar digestion of α-syn. All reactions were carried out in a total volume of 500 µL reaction buffer (50 mM NaOAc, 20 mM NaCl, 5 mM DTT, pH 5) at 37 °C and agitated at 600 rpm.

## Fibril formation and ThT kinetics for 66–140 and mutants
Prior to aggregation, protein samples were exchanged into pH 7.4 buffer (20 mM NaPi, 140 mM NaCl) using Zeba spin columns (Thermo Scientific) and filtered through YM-100 spin units (Millipore) to remove any preformed aggregates. ThT-monitored aggregation reactions (70 µL, [protein] = 30 – 80 µM, 0.01% $NaN_3$, [ThT] = 10 µM for 30 and 40 µM protein and 20 µM for 80 µM protein) were carried out in sealed black, polypropylene, 384-well flat-bottom microplates (781209, Greiner Bio-one) supplemented with 2-mm glass beads. Post-aggregation, seeds (66–140, K80A, E83A, K96A, K97A, D98A and K102A) were prepared by ultracentrifugation (434,902 × *g*, TLA100.2 rotor, Beckman Coulter, for 45 min at 4 °C). Pelleted seeds were quantified by absorbance measurements after dissolution by 3 M guanidinium hydrochloride. Seeds (0.2–1 µL, 100 µM) were added to a solution of either 30 µM 66–140 or mutant protein, to a final volume of 70 µL in the absence of 2 mm glass beads. ThT fluorescence (excited and monitored at 415 and 480 nm, respectively) was recorded at 37 °C with continuous linear shaking (1 mm, 1440 rpm) using a microplate reader (Tecan Infinite M200 Pro). Fibril samples for Raman ([protein] = 60 – 214 µM) and ssNMR ([protein] = 145 µM) measurements were aggregated under similar conditions in the absence of ThT and prepared by ultracentrifugation.

At least 2 independent experiments were performed with at least 4 replicates for each condition on each plate. For UV analysis of soluble and insoluble fractions after aggregation, separation was achieved by ultracentrifugation (434,902 × *g*, TLA100 rotor, Beckman Coulter) for 45 min at 4 °C. The pelleted protein was quantified by absorbance measurements after dissolution by 3 M guanidinium hydrochloride. Kinetics data are plotted in IgorPro 9.04.

## Fibril formation and ThT kinetics for Ac1−140 and mutants

Prior to aggregation, protein samples were exchanged into pH 7.4 buffer (20 mM NaPi, 140 mM NaCl) using Zeba spin columns (Thermo Scientific) and filtered through YM-100 spin units (Millipore) to remove any preformed aggregates. ThT-monitored aggregation reactions (70 μL, [protein] = 50 and 100 μM, 0.01% $NaN_3$, [ThT] = 10 and 20 μM) were carried out in sealed black, polypropylene, 384-well flat-bottom microplates (781209, Greiner Bio-one) supplemented with 2-mm glass beads. Fibril seeds (Ac1−140, AcK80A, AcK96A and AcK97A) were prepared by aggregating at 100 μM in the absence of ThT and pelleted by ultracentrifugation (434,902 × $g$, TLA100.2 rotor, Beckman Coulter, for 45 min at 4 °C). Pelleted seeds were quantified by absorbance measurements after dissolution by 3 M guanidinium hydrochloride. Seeds (1.7–3.5 μL, 50–100 μM) were added to a solution of either 50 or 100 μM Ac1−140 or mutant protein, to a final volume of 70 μL in the absence of 2 mm glass beads. ThT fluorescence (excited and monitored at 415 and 480 nm, respectively) was recorded at 37 °C with continuous linear shaking (1 mm) using microplate readers (Tecan Infinite M200 Pro and Spark). Fibril samples ([protein] = 100 μM) for Raman measurements were aggregated in eppendorfs in the absence of ThT and prepared by ultracentrifugation.

## Proteinase K degradation of α-syn fibrils

In Eppendorf 1.5 mL tubes, 66–140 and mutant fibrils (30 μM) were incubated with Proteinase K (Invitrogen) at different concentrations (2 and 0.2 ng) in reaction buffer (20 mM NaPi, 140 mM NaCl, pH 7.4) in a total volume of 50 μL. Samples were agitated at 600 rpm for 20 h at 37 °C in a Mini-Micro 980140 shaker (VWR). Reactions were terminated with 0.1% TFA and 3 M guanidinium hydrochloride.

## LC-MS

Samples (2–5 μL) in 0.1% TFA and 3 M guanidinium hydrochloride were separated using a HPLC (Agilent 1100 series HPLC, Agilent Technologies) on a reversed-phase C18 column (Zorbax, 2.1 × 50 mm, 3.5 μm, Agilent Technologies). Peptides masses of α-syn were obtained with an Agilent 6230 electrospray ionization time-of-flight LC-MS. For mobile phase, a gradient (0–50% acetonitrile and 0.05% TFA) at a flow rate of 0.2 mL/min was used. The HPLC systems and MSD were controlled, and data analyzed using the Agilent MassHunter Workstation platform. Mass spectra were obtained using positive-ion mode. Peptide identification from experimental masses was determined using Expasy FindPept [https://web.expasy.org/findpept/] with a mass tolerance of 1 dalton. Two independent experiments were analyzed. Results are shown in Supplementary Tables 1–3, 6 and 8.

## TEM

Samples (10 μL) were applied to TEM grids (400-mesh formvar and carbon coated copper, Electron Microscopy Sciences) for approx. 2 min and wicked away by filter paper. Deionized water (10 μL) was then applied and wicked away immediately. A solution of 1% uranyl acetate (10 μL) is placed on the grid for 2 min, wicked away, and air-dried. TEM was performed using either a JEOL JEM 1200EX (NHLBI EM Core Facility) equipped with an AMT XR-60 digital or JEM 1400EX transmission electron microscope equipped with an AMT Nanosprint 43 MKII camera. Fibril helical twists were calculated using Fiji.

## AFM

Aminopropyl-silatrane (APS) solution was used to modify the mica substrates prior to deposition (5 μL, ~10 nM protein). Protein samples were incubated for 10 min and then gently washed with 200–300 μL deionized water followed by drying with $N_2$. Dried samples were imaged using the Multimode-8 AFM (Bruker-Nano, Santa Barbara, CA) in the oscillating (tapping) mode, using silicon probes with nominal stiffness of 2.8 N/m and resonance of 70 kHz (FESP, Bruker-nano, Santa Barbara, CA). Final scan sizes were 3–5 μm, at resolutions of ~1 nm/

pixel. Oscillation amplitudes were adjusted to about 5–6 nm, and the setpoint amplitude was set to about 90% of nominal. Acquired images were preprocessed using the instrument software (Nanosope Analysis 2.0) and the imaged fibrils were traced along their mid-lines using ImageJ and a semi-automating code written in Matlab. Fibrils from each construct ($n = 11$–31) were analyzed, and the mean and standard deviation for each wavelength were calculated. For individual fibrils, the Fourier peak was fitted with a Gaussian whose width (±σ) measures wavelength variability within the fibril. For most fibrils, the standard deviation was in the range of 2–5% of the period, and it never exceeded 10% of the period in any fibril.

## Raman spectroscopy

Samples (20 μL) were plated in 18-well #1.5H cover glass chambers (CellVis) overnight in a humidity chamber prior to measurements. Raman spectroscopy was conducted as previously reported[72,73] with some modifications. Excitation at 488-nm (Coherent Sapphire-SF, 80 mW at the sample) was directed to the sample through a laser clean-up filter (LL01-488-25, Semrock), a dichroic mirror (Di02-R488-25 × 36, Semrock), and a UPlanSApo 60×/1.30 NA silicon oil objective (Olympus). Raman scattering was collected by the same objective and passed through a long-pass filter (BLP01-488R-25, Semrock). Spectra were collected with a 1200 mm$^{-1}$ grating for 10 accumulations with an integration time of 2-s (Horiba Symphony II liquid-nitrogen cooled back-illuminated deep-depletion CCD detector, 20 kHz, best dynamic range, pixels 124 to 132 were binned in the $y$-dimension). Calibration was performed using neat cyclohexane. A constant offset was subtracted from all spectra, and buffer subtraction was accomplished using LabSpec 6 software (Horiba Scientific). Spectra were normalized to the Phe breathing mode (1003 cm$^{-1}$) using Matlab R2022b (Mathworks) for comparison. At least two independent aggregation reactions were collected per construct with at least 7 individual measurement locations.

## ssNMR

ssNMR measurements were performed at 17.6 T using a Tecmag Redstone spectrometer and a NMR probe constructed in the laboratory of Dr. Ago Samoson (Tallinn University of Technology, Estonia). Uniformly $^{13}$C, $^{15}$N-labeled 66–140 fibrils were pelleted and lyophilized, and packed into a 1.8 mm MAS rotor (Revolution NMR, Inc) as a powder. Once packed, the fibril powder was minimally rehydrated with a pH 7.4 buffer (4 mM NaPi, 28 mM NaCl). Total sample volume was about 8 μL. $^1$H-$^{13}$C/$^{15}$N polarization transfers were carried out using cross polarization (CP), which selects only signals from immobilized regions of the samples, and is not sensitive to flexible, rapidly moving portions of the protein. $^{15}$N-$^{13}$C polarization transfers and $^{13}$C-$^{15}$N polarization transfers were also carried out using CP. $^{13}$C-$^{13}$C polarization transfers were carried out with a 25–25 ms DARR/RAD period[74]. Additional experimental details, including pulse sequence parameters and total measurement times, are given in Supplementary Table 4. Pure Gaussian apodization functions were used to process all data.

ssNMR signal assignments are presented in Supplementary Table 5. Spectra were assigned using a computational Monte Carlo/simulated annealing (MCSA) algorithm[53] which takes as inputs the backbone amide $^{15}$N, carbonyl $^{13}$CO, and $^{13}$Cα frequencies acquired in 2D/3D NCaCX, NCOCX, and CaNCOCX experiments, and the protein sequence.

Quantitative $^{13}$C-$^{15}$N distance measurements were carried out using $^{13}$C-detected frequency-selective rotational-echo double resonance (FS-REDOR) experiments, using the pulse sequence described by Jaroniec et al.[75]. Two 15 μs 180° pulses per rotor period were applied to $^{15}$N during the REDOR dephasing period. Gaussian 180° pulses 0.33 ms long were applied simultaneously to both $^{13}$C and $^{15}$N spins half-way through the REDOR dephasing period $t_{REDOR}$. The selective $^{13}$C 180° pulse was applied on-resonance with the E83 sidechain $^{13}$C$_\delta$ signal

(Supplementary Fig. 13), and inverts only the signals around 183.9 ppm, consisting of E83 and Q79 sidechain $^{13}C_\delta$ and the backbone $^{13}CO$ signals. The selective $^{15}N$ 180° pulse was centered on the lysine sidechain $^{15}N$ signals (Supplementary Fig. 13), and inverts only these peaks. Dephased spectra $S_1$ and un-dephased reference spectra $S_0$ were recorded as a function of $t_{REDOR}$ with and without the selective $^{15}N$ 180° pulse, respectively. The fractional difference signal $(S_0-S_1)/S_0$ shows the amount of dephasing due to $^{13}C$-$^{15}N$ dipolar coupling arising between E83 sidechain $^{13}C_\delta$ and lysine sidechain $^{15}N$. XY16 phase cycling was applied to the $^{15}N$ REDOR pulses. The measured dephasing curve is reproduced by quantum mechanical density matrix simulations which included the finite 15 μs 180° $^{15}N$ pulse lengths, the effects of XY16 phase cycling, and the 12 kHz MAS, assuming a 3.2 Å $^{13}C$-$^{15}N$ distance. Simulations assumed an "effective $T_1$" for $^{15}N$ spins caused by a combination of dephasing during $^{15}N$ pulses and imperfections in $^{15}N$ 180° pulses, which attenuates the REDOR signal as a function of $t_{REDOR}$[76].

## Molecular dynamics simulations

Model 8, containing two protofibrils made up of five protomers each, of the NMR structure of α-syn (PDB ID 8FPT) was modified using Chimera (UCSF)[77]. The phi- and psi-angles of residues 69, 77, and 92 were adjusted to the values reported in Supplementary Table 5 to ensure the Kabsch and Sander algorithm[78] would calculate the closest β-sheet assignments to those determined experimentally. The protein model was solvated by a periodic 113 Å cubic box of TIP3 water, Na$^+$ counter ions were added to neutralize the charge of the system, and additional Na$^+$ and Cl$^-$ ions were added to bring the system to 140 mM NaCl. The system contained a total of 50,602 atoms, consisting of 15,044 water molecules, 135 Cl$^-$ atoms and 115 Na$^+$ atoms. The energy of the system was minimized using the steepest descent algorithm to remove unfavorable interactions. The energy-minimized system was equilibrated for 100 ps while the temperature was increased from 0 to 300 K. A 30-ns production MD was run starting from the equilibrated structure (Supplementary Fig. 25a) using GROMACS on the NIH HPC Biowulf Cluster with a CHARMM36m force field[78]. Based on the root mean square deviation, the last 15 ns of the resulting trajectory was analyzed using VMD[78] and Chimera (UCSF)[77].

## Reporting summary

Further information on research design is available in the Nature Portfolio Reporting Summary linked to this article.

## Data availability

All data supporting the results of this study can be found in the article, supplementary information, source data file, and Figshare [https://doi.org/10.25444/nhlbi.28513553]. The source data, primer list, and gel images have been compiled and provided as source data files for this paper. The structural data used for MD in this are available in the PDB database under assession code 8FPT. NMR chemical shifts are available in the BMRB database (ID 52983). [https://bmrb.io/data_library/summary/index.php?bmrbId=52983] Source data are provided in this paper.

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

## Acknowledgements

This work was supported by the Intramural Research Program at the NIH, the National Heart, Lung, and Blood Institute (NHLBI), the National Institute of Biomedical Imaging and Bioengineering, and the National Institute of Diabetes and Digestive and Kidney Diseases. Support was provided to JCL under project ZIA-HL001055. LC-MS and TEM (JOEL JEM1200EX) were performed on instruments maintained by the NHLBI Biochemistry and EM Core, respectively. We thank Kem Sochacki for training to use the JOEL 1400 microscope and the Taraska Lab (NHLBI) for maintaining the instrument. We thank Robert Tycko for useful discussions and help with ssNMR measurements. We thank Nahid Tayebi and Ellen Sidransky for the generous gift of dissected mouse brains (Medical Genetics Branch, National Human Genome Research Institute, NIH). pT7-7 α-syn was a gift from Hilal Lashuel (Addgene plasmid #36046), and pNatB was a gift from Dan Mulvihill (Addgene plasmid #53613). We thank Jared Shadish for performing PCR in making the K80A, K96A, and K97A full-length α-syn plasmids. This work utilized the computational resources of the NIH HPC Biowulf cluster (https://hpc.nih.gov).

## Author contributions

R.P.M. and J.C.L. conceived the project and wrote the manuscript with contributions from all authors; R.P.M. and J.C.L. made protein samples; R.P.M. collected aggregation kinetics and TEM images, and performed PK-digestion and LC-MS experiments; S.R. collected and analyzed Raman data and performed MD simulations; E.K.D. collected and analyzed AFM data; C.B.W. collected and analyzed ssNMR data.

## Funding

## Competing interests

The authors declare no competing interests.
