## [Transparent Peer Review file · Nature Communications]

Defining essential charged residues in fibril formation of a lysosomal derived N-terminal α -synuclein truncation

Corresponding Author: Dr Jennifer Lee

Version 0:

Reviewer comments:

Reviewer #1

(Remarks to the Author)

McGlinchey et al. used ThT assays, TEM, Raman spectroscopy, and ssNMR to discover new electrostatic interactions and new salt bridge pairs in recombinant alpha-synuclein. They performed mutational studies to characterize individual residues that are critical for fibril polymorphism and elongation. They performed mutational studies on segment 66-140 and showed charge neutralization effects on aggregation kinetics for K80, E83, K96 and K97 mutants. They used Raman spectroscopy discovering that K80 and E83 are critical for defining the amyloid structure of segment 66–140. And they performed charge-switch mutation studies to test three potential salt-bridge pairs. McGlinchey et al.'s work suggests the importance of electrostatic interactions of residues in fibril formation, propagation and polymorphism of recombinant alpha-synuclein, and they discovered a new salt-bridge pair of E83–K97.

The variety of methods applied to study electrostatic interactions in this protein is impressive; the results are solid, and support the authors' claims. Nevertheless, the novelty of the work and the advance in knowledge to the field are quite limited.

1) The relevance of this work to pathological asyn fibril structures is uncertain. Currently cryoEM structures of patient brain-extracted alpha-synuclein fibril structures from MSA, DLB, PD, and JOS have been determined. And notably recombinant asyn fibril structures only share partial structural similarities with MSA asyn fibril core, and almost none with PD, DLB or JOS. The observation that all brain fibril structures showed single morphology (except for minor structural changes for MSA I and MSA II) but wildtype recombinant asyn showed two polymorphs indicates that recombinant samples cannot be simply correlated to human brain samples. For example, the potential new salt bridge pair E83-K97 are not observed in any of the human brain fibril structures. The present work's significance will be greatly increased if any of the hypotheses can be tested on recombinant fibrils that have the same fibril core as human asyn (only recombinant tau has fully captured the AD structure so far), or the hypotheses can somehow be tested on human fibril samples.

2) The importance of K80, E83 has been previously well studied. Many papers have studied the salt bridge formed by E46-K80. The paper by Zhou et al. in 2022 titled "N-homocysteinylation of alpha-synuclein promotes its aggregation and neurotoxicity" showed that modifications on K80 affect asyn's aggregation and seeding activities. The paper by Waxman et al. in 2010 titled "Residue Glu83 plays a major role in negatively regulating asyn amyloid formation" has also studied the importance of E83.

McGlinchey et al.'s work overall supports their conclusions and claims.

Minor suggestions:

1) For Figure 1D, the spectrum seems to have an overall higher intensity for Ac1-140 than 66-140, and the sharpened amide-III and narrower amide-II bands are not obvious to me. It will be helpful if a difference spectrum provided like Figure 4.

2) In results section "K80, E83, and K97 are key residues for fibril propagation", the last paragraph contains several passages that are essentially "discussions" of the results (For example, starting from "Likewise, the spectrum of cross-seeded..."). I suggest moving these passages to the "Discussion" section.

Reviewer #2

(Remarks to the Author)

I co-reviewed this manuscript with one of the reviewers who provided the listed reports. This is part of the Nature Communications initiative to facilitate training in peer review and to provide appropriate recognition for Early Career

Researchers who co-review manuscripts.

Reviewer #3

(Remarks to the Author)

Jennifer Lee and co-workers present a study of an alpha-synuclein truncation and mutants in order to gain insights into the roles of various charged residues in the kinetics of fibril growth and structural polymorphism. The study overall is of high quality, with excellent experimental results and voluminous amounts of data. Unfortunately, this reviewer concludes that the study falls short of the level of clarity in the conclusions required for publication as a communication. These points are elaborated in the comments below.

1. There is a concern about the construct used, that the case for 66-140 is not made very convincingly.

The authors state "Specifically, we chose a pathologically relevant N-terminally truncated variant, 66–140 (33) for investigation." Only one citation is included here, to Ref. 33, which (after re-reading it) this reviewer did not find a significant point of emphasis on the 66-140 truncation in that study. Although some underlying data may be present there, it is not a major point of emphasis and certainly does not explicitly address this construct in the main text. Some more recent papers make reference to finding 66-140 in tissue, but the authors did not cite further examples or explain the significance. It seems very likely that in tissue the N terminus is truncated after fibrils form, so the relevance of studying the truncation in vitro are not well argued.

Further, the authors note that there are many charged residues in alpha-syn, but this construct removes the majority of Lys residues, greatly reducing the number of potential salt bridges that could form. It becomes apparent after reading the full paper that this might be the intentional choice for the design of the experiments, but this needs to be more clearly explained in the introduction.

Although it is true that "nearly all WT alpha-syn fibrils contain at least one salt bridge", most of these involve Lys residues within the region 1-65 among the KTKEGV imperfect repeats.

So overall, the choice of construct certainly reduces the significance and generality of the results for disease relevance in this reviewer's opinion. Nevertheless, the study could still be interesting from the perspective of protein biophysics.

2. The data overall are high quality but there are many confounding observations, so the hypothesis(es) are not clearly stated and the experimental measurements, although voluminous, do not seem to be focused on testing hypotheses in a way that arrives at satisfying conclusions. For example from the discussion and conclusions there are several statements along these lines

"While it was expected that Ala-substitutions at K80 and E83 would have strong influences as they are found within the ssNMR structure core, the dramatic effects of K96A and K97A were not as they reside outside in the disordered region."

"Inefficiency of fibril propagation of the native 66–140 structure by E83A makes sense because E83A forms a distinct fibril polymorph which is incompatible with 66–140. For K80A, the results are conflicting."

"For K97A, its behavior is puzzling because the data would suggest that it adopts a very similar structure, and its able to cross-seed soluble 66–140"

Overall, it seems that there are many cases where the seeding, cross-seeding, and structural features do not follow a consistent pattern in the data. Or if there is a pattern, it is indirect.

3. Another problematic issue is the discussion of salt bridge measurements comparing the NMR and Raman results. Initially it is noted that Raman bands are more ordered in 66-140 than Ac1-140, such as sharpened amide III (1226 cm⁻¹) and narrower amide-II (1560 cm⁻¹) bands. In addition there is a new peak feature at 1012.5 cm⁻¹, which eventually in the discussion is mentioned as a possible salt bridge feature, but when first raised, there is no attribution to a potential structural feature correlated with the spectral fingerprint. It would be helpful at minimum to be explicit at this point of the manuscript and indicate that the 1012.5 cm⁻¹ is postulated to be due to a new salt bridge geometry or environment.

But this is hard to follow since later the authors point out that there is no prior evidence of similar peaks in alpha-syn fibril studies, even though many of them are known to have salt bridges. Moreover, although there is a REDOR NMR measurement to indicate that E83 is involved in a salt bridge, the potential assignment of K97 is not a resolved residue in the NMR assignments, which end at 95. Then a series of cross-seeding experiments muddy the waters further, for example where substitutions at K97 do not have the expected effects consistent with an essential salt bridge. So then the argument shifts to K97 potentially having an indirect role. But then how can this be a salt bridge in the structure by NMR and explain the 1012.5 cm⁻¹ peak?

In the end, these are all very interesting observations, and there is the foundation for a potentially impactful study, but as it stands, this manuscript has too many uncertainties and open ends to be ready for publication as a high impact communication.

Reviewer #4

(Remarks to the Author)

The authors investigated the role of charged residues in the 66-140 segment of α -synuclein on the aggregation pattern. Morphology of fibrillar aggregates assembled by the wild type 66-140 and asset of mutants was characterized by AFM and TEM. In parallel, the structure of aggregates was probed by NMR and Raman spectroscopies. The major goal of the described experiments was to elucidate the polymorphism of fibrils formed by α -synuclein, but the authors fall short in accomplishing the goal as truncated form of the full-length α -synuclein are different from the ones assembled by the truncated protein. Although replacing charged residues with alanine facilitate the aggregation kinetics, the extension of the described approaches to the full-length α -synuclein is needed to make the results biologically significant. Additionally, there are multiple serious weaknesses in the paper that are listed below.

1. Structure of aggregates and its relation to the morphology of fibrils. Structurally the aggregates were characterized by NMR and Raman spectroscopies and these studies revealed the formation of characteristic structural features of the aggregates such as β -sheets assigned to specific location in the mild type 66-140 protein. However, the link of these structural features with morphologies of fibrils revealed by TEM and AFM imaging is questionable. The yield of fibrils was considerably less than 100%, so the comparison of imaging and spectroscopic data could be made if only fibrils were used for the secondary structure analysis. This concern is critical when similar studies were made for mutants, which were characterized the low yield of fibrils. For example, the yield of fibrils was as low as 13% for K97A mutant.

2. Cross-seeding experiment were used to reveal the differences is structures of fibrils assembled by mutants and the wild-type 66-140 sample. To eliminate the contribution of aggregates formed the wild-type 66-140 sample in the aggregation process, the authors used the approach at which aggregation process of the wild-type 66-140 sample is fully blocked. The approach is neither described nor explained, so one could assume that the difference in cross-seeding experiments is due to unexplained features of the aggregation kinetics rather than the blocking the seeding reaction of the wild-type 66-140 aggregation by non-homologous structure of fibrils seeds of the mutant (E83A). The approach should be described and its potential contribution of the aggregation process should be carefully discussed.

3. Morphologies of fibrils was studied by AFM and TEM. The selected images are of the high quality, but no analysis of the variability between such feature of twisted fibrils as their periodicity is provided. The periodicity numbers are indicated, but no explanation on the statistics of these measurements is provided. Unbiased analysis is needed to support the numbers on the fibril's periodicity listed in the paper. What is the partition between different morphologies for each protein? What is the variability of the periodicity along the selected fiber, so the morphological heterogeneity was observed for individual fiber? Additional unanswered questions: Are the individual fibrils have only one periodicity or there are segments within each fibril with two different pitch values? How does the periodicity including the partition between the various types vary for different mutants? How does this feature translate into the seeded fibrils? How close are the data obtained with TEM and AFM approaches?

4. The link of mutations with the sites of potential salt bridges in aggregates formed by the wild-type 66-140 is discussed but experiments at different ionic strengths could provide the data enabling the authors to test the major hypothesis on the role of the salt bridges in the aggregation process.

Version 1:

Reviewer comments:

Reviewer #2

(Remarks to the Author)

We believe the authors have addressed our concerns in the revised manuscript. However, we still think the manuscript lacks novelty so our suggestion remains the same (do not recommend for publication in Nat.Com.)

Reviewer #3

(Remarks to the Author)

The authors have thoroughly addressed the concerns raised in my review. I believe the study is now suitable for publication, pending the option of other reviewers' concerns.

Reviewer #4

(Remarks to the Author)

The authors responded to my comments.

We would like to thank all Reviewers for their careful reading and thoughtful comments and suggestions to improve our manuscript. We have addressed Reviewers' concerns with new experimental data with the addition of two new main figures (1 and 8) and SI figures (1–3, 6, 25–28) and tables (1–3). The manuscript has been significantly revised. Annotated versions of the manuscript and supporting information with changes marked in blue has been uploaded as Supporting Information for Review Only. Point-by-Point responses follow.

Response to Reviewer 1 (*Reviewer comments are reproduced in black and authors' responses are in blue*)

Reviewer #1 (Remarks to the Author):

McGlinchey et al. used ThT assays, TEM, Raman spectroscopy, and ssNMR to discover new electrostatic interactions and new salt bridge pairs in recombinant alpha-synuclein. They performed mutational studies to characterize individual residues that are critical for fibril polymorphism and elongation. They performed mutational studies on segment 66-140 and showed charge neutralization effects on aggregation kinetics for K80, E83, K96 and K97 mutants. They used Raman spectroscopy discovering that K80 and E83 are critical for defining the amyloid structure of segment 66–140. And they performed charge-switch mutation studies to test three potential salt-bridge pairs. McGlinchey et al.'s work suggests the importance of electrostatic interactions of residues in fibril formation, propagation and polymorphism of recombinant alpha-synuclein, and they discovered a new salt-bridge pair of E83–K97.

The variety of methods applied to study electrostatic interactions in this protein is impressive; the results are solid, and support the authors' claims. Nevertheless, the novelty of the work and the advance in knowledge to the field are quite limited.

REPLY: We thank the reviewer for their positive response on the quality of our data. In addressing the concerns of novelty and advancement in knowledge, we have determined that a specific lysosomal protease, asparagine endopeptidase (AEP), is responsible for the generation of 66–140 by using lysosomes isolated from mice brains. Despite being previously identified from PD patients, the origin of 66–140 was unknown. AEP is pathologically relevant as it is upregulated in PD. The inclusion of this data also prompted us to change the title and revise the introduction to elaborate on the broader biological and pathological relevance of α -synuclein truncation generation by the lysosome. Importantly, we have also directly observed and assigned 8 out of the 13 previously reported PD-related truncations to lysosomal proteases for both N- and C-terminal truncations. We also show that 66–140 is derived from degradation of soluble, and not fibrillar α -synuclein, which warrants a detailed investigation of its aggregation propensity. In strengthening the connection to disease, we also assessed lysosomal protease activity in mice as a function of age. Enriched brain lysosomes were isolated from mice euthanized at 2 vs. 17 months of age, and showed older mice had elevated protease activity levels. These enhanced activities strongly correlate in promoting α -synuclein truncations. The new **Figs. 1, S1, and S2** are reproduced below for your convenience. Overall, we feel strongly that this additional work adds biological significance in studying this specific construct. We hope that these additional experiments will be sufficient to satisfy the Reviewer.

Figure 1. Lysosomal degradation of α -syn. **(A)** Schematic representation of the primary amino acid sequences of 1–140 (top) and 66–140 (bottom), coloring basic (blue) and acidic (red) residues. Charged residues that participate in salt bridges within the shared α -syn fibril core (residues 36–99) are indicated^{3, 4}. **(B)** Schematic representation of the primary amino acid sequence of α -syn with cleavage sites generated for either soluble (cyan) or fibrillar (black) α -syn. Residues 36–99 (light gray) show cryoEM fibril core while residues 61–95 denote the NAC region (dark gray). **(C)** α -Syn peptide fragments derived from lysosomal degradation of soluble (cyan) and fibrillar (black) α -syn. Fragment masses and residue assignments are reported in **Table S1**. Previously identified fragments from PD patients are denoted by asterisks. Effect of age on specific protease activities in mouse brain lysosomal extracts from 2 and 17 months of age. Fluorogenic substrates **(D)** Ac-RR-AMC for CtsB, **(E)** Ac-FR-AMC for CtsL, **(F)** MCA-GKPILEFRK(L-Dnp)-D-R-NH₂ for CtsD and **(G)** AENK-AMC for AEP were incubated with lysosomal extracts (10–40 μ g total protein) at pH 5.0 with 5 mM DTT, 37 °C. Fluorescence was recorded as a function of time (30 and 60 min) and relative fluorescence units (RFU) are reported ($n = 3$).

Figure S1. CtsB, CtsL, AEP and CtsD degradation of soluble α -syn. SDS-PAGE analysis of CtsB (A), CtsL (B), AEP (C) and CtsD (D) degradation of soluble α -syn over time at pH 5. (E) Peptide fragments identified by LC-MS analysis from CtsB (cyan), CtsL (green), AEP (black) and CtsD (purple) activity of soluble α -syn. (F) Schematic representation of the primary amino acid sequence of α -syn (residues 1 to 140) with LC-MS-mapped cleavage sites for CtsB (cyan), CtsL (green), AEP (black) and CtsD (purple) generated from soluble α -syn.

Figure S2. CtsB, CtsL, AEP and CtsD degradation of fibrillar α -syn. **(A)** Peptide fragments identified by LC-MS analysis from CtsB (cyan), CtsL (green), AEP (black) and CtsD (purple) activity on fibrillar α -syn. **(B)** Schematic representation of the primary amino acid sequence of α -syn (residues 1 to 140) with LC-MS-mapped cleavage sites for CtsB (cyan), CtsL (green), AEP (black) and CtsD (purple) generated from fibrillar α -syn.

1) The relevance of this work to pathological asyn fibril structures is uncertain. Currently cryoEM structures of patient brain-extracted alpha-synuclein fibril structures from MSA, DLB, PD, and JOS have been determined. And notably recombinant asyn fibril structures only share partial structural similarities with MSA asyn fibril core, and almost none with PD, DLB or JOS. The observation that all brain fibril structures showed single morphology (except for minor structural changes for MSA I and MSA II) but wildtype recombinant asyn showed two polymorphs indicates that recombinant samples cannot be simply correlated to human brain samples. For example, the potential new salt bridge pair E83-K97 are not observed in any of the human brain fibril structures. The present work's significance will be greatly increased if any of the hypotheses can be tested on recombinant fibrils that have the same fibril core as human asyn (only recombinant tau has fully captured the AD structure so far), or the hypotheses can somehow be tested on human fibril samples.

REPLY: We acknowledge the Reviewer's questioning in regards to the relevance of this work to pathological α -syn fibril structures. Currently, we believe that these experiments are not yet feasible,

and we believe this is outside the scope of this work. However, we do appreciate the interest in applying these hypotheses to human fibril samples. Until experimental solution conditions have been established to show faithful propagation of human fibril polymorphs, the Reviewer's concern cannot be addressed. To recognize and bring forth this deficiency, we have included a sentence in the Conclusion, acknowledging this need as well as the current challenges amplifying the structures of patient-derived material. Furthermore, in trying to strengthen the manuscript in this vain, we have added a section in the Introduction to substantiate the relevance of α -synuclein truncations in disease.

That said, in also addressing suggestions by Reviewer 4, who was more interested in applying these studies to the full-length protein, we did generate and characterize three acetylated (Ac) AcK80A, AcK96A and AcK97A mutants of the full-length α -syn. These data shown in **Figure 8** along with supporting **SI Figures S26–S28**, which are included below for your convenience. In summary, both AcK96A and AcK97A mutants mirror the sluggish behaviors of the corresponding 66–140 mutants. Interestingly, Ala-substitution of K80 had opposite behaviors for full-length and 66–140, where aggregation kinetics is accelerated and slowed, respectively. These results support that 66–140 as a useful model for investigating α -syn fibril formation, offering relevant sites that modulate aggregation propensity.

Figure 8. Aggregation, structural characterization and cross-seeding kinetics of single-Ala mutants of Ac1–140. **(A)** Schematic representation of the primary amino acid sequences of Ac1–140 showing sites of mutations. CryoEM fibril core assigned to residues 36 to 99. Aggregation reactions monitored by ThT fluorescence (10 μ M) of 50 μ M Ac1–140 **(B)**, AcK80A **(C)**, AcK96A **(D)** and AcK97A **(E)** in pH 7.4 buffer (20 mM NaPi, 140 mM NaCl) at 37 $^{\circ}$ C with continuous linear shaking supplemented with a 2-mm glass bead ($n \geq 4$). **(F)** Fibril morphologies of single-Ala mutants visualized by TEM. Scale bars are 100 nm. Full-size images are shown in **Fig. S27**. **(G)** Aggregation kinetics monitored by ThT fluorescence (10 μ M) of soluble Ac1–140 (50 μ M) seeded with 2.5 μ M Ac1–140 (black), K80A (blue), K96A (purple) and K97A (green) fibrils in pH 7.4 buffer (20 mM NaPi, 140 mM NaCl) at 37 $^{\circ}$ C with continuous linear shaking. Unseeded 66–140 (black) is also shown as a control. Solid lines and shaded regions represent mean and SD, respectively ($n \geq 5$).

Figure S26. Aggregation kinetics monitored by ThT fluorescence of Ac1-140 (*black*), K80A (*cyan*), E83A (*magenta*), K96A (*purple*) and K97A (*green*), protein (100 μ M) in pH 7.4 buffer (20 mM NaPi, 140 mM NaCl) at 37 °C with continuous linear shaking supplemented with a 2-mm glass bead ($n \geq 5$).

Figure S27. Representative TEM images of Ac1-140, AcK80A, AcK96A and AcK97A fibrils. Dashed areas represent images used in **Fig. 8F**. Scale bars are as shown.

Figure S28. (A). Full Raman spectra of Ac1-140 and single Ala mutants. Solid lines and shaded regions represent mean and SD, respectively ($n \geq 5$). Dashed lines represent areas of interest. (B) Expanded view of the fingerprint (left) and Amide I (right) regions from the spectra in panel A. (C) Difference spectra generated by subtracting the average mutant from the average Ac1-140 spectrum. The error has been propagated from the SD of the averaged spectra.

2) The importance of K80, E83 has been previously well studied. Many papers have studied the salt bridge formed by E46-K80. The paper by Zhou et al. in 2022 titled “N-homocysteinylation of alpha-synuclein promotes its aggregation and neurotoxicity” showed that modifications on K80 affect asyn’s

aggregation and seeding activities. The paper by Waxman et al. in 2010 titled “Residue Glu83 plays a major role in negatively regulating asyn amyloid formation” has also studied the importance of E83.

REPLY: We thank the Reviewer for bringing these papers to our attention. After reading both papers, we deemed that these data are not relevant. We respectfully disagree that these study contribute to our understanding on the importance of K80 and E83 in α -syn fibril formation. Firstly, the paper by Zhou *et al.* uses largely mouse and HEK-293T cells for study. It is not evident that the chemical modification was specific to K80 on recombinant protein and caused a direct effect on in vitro aggregation. It was inferred by a K80R mutant. Moreover, a K80-homocysteinylation mutant which is a significant residue modification and likely causes major structural changes to fibril formation. So, this does not simply address a charge effect. The other paper reports using an E83A mutant on non-aggregating Δ 74–79 and Δ 71–82 constructs. So, this study is not relevant and does not apply to the WT protein. Furthermore, to the best of ability, we could not find any other studies that report detailed understanding into the role(s) of K80, E83, K96 and K97 in fibril formation.

McGlinchey et al.’s work overall supports their conclusions and claims.

Minor suggestions:

1) For Figure1D, the spectrum seems to have an overall higher intensity for Ac1-140 than 66-140, and the sharpened amide-III and narrower amide-II bands are not obvious to me. It will be helpful if a difference spectrum provided like Figure4.

REPLY: As suggested, this is added as a new **SI Figure 6** shown below.

2) In results section “K80, E83, and K97 are key residues for fibril propagation”, the last paragraph contains several passages that are essentially “discussions” of the results (For example, starting from “Likewise, the spectrum of cross-seeded...”). I suggest moving these passages to the “Discussion” section.

REPLY: As recommended, we have removed any discussion points in the Results and moved them to the Discussion.

Reviewer #2 (Remarks to the Author):

REPLY: We thank the Reviewer for their time and have provided detailed responses to Reviewer 3.

Reviewer #3 (Remarks to the Author):

Jennifer Lee and co-workers present a study of an alpha-synuclein truncation and mutants in order to gain insights into the roles of various charged residues in the kinetics of fibril growth and structural polymorphism. The study overall is of high quality, with excellent experimental results and voluminous amounts of data. Unfortunately, this reviewer concludes that the study falls short of the level of clarity in the conclusions required for publication as a communication. These points are elaborated in the comments below.

REPLY: We thank the Reviewer for the positive comments on data quality. To address Reviewer's concern, we have made substantive changes to our manuscript to elevate the novelty and significance of the study by adding new data addressing (1) the biological origin of 66–140 from the soluble state of full-length α -syn (**Figs. 1, S1, S2** and **Tables S1–S3**), (2) the presence of a E83-K97 salt bridge by molecular dynamics simulations modeling (**Fig. S25**), and (3) extending K-to-A mutations to the full-length protein (**Figs. 8** and **S26–S28**). We have revised our writing for clarity. We hope that these efforts will convince the Reviewer to recommend our manuscript for acceptance. The new figures are reproduced on pages 2–7 in the response letter.

1. There is a concern about the construct used, that the case for 66-140 is not made very convincingly.

The authors state "Specifically, we chose a pathologically relevant N-terminally truncated variant, 66–140 (33) for investigation." Only one citation is included here, to Ref. 33, which (after re-reading it) this reviewer did not find a significant point of emphasis on the 66-140 truncation in that study. Although some underlying data may be present there, it is not a not a major point of emphasis and certainly does not explicitly address this construct in the main text. Some more recent papers make reference to finding 66-140 in tissue, but the authors did not cite further examples or explain the significance. It seems very likely that in tissue the N terminus is truncated after fibrils form, so the relevance of studying the truncation *in vitro* are not well argued.

REPLY: We agree with the Reviewer that a better job could have been done to support the choice of the 66–140 construct and its biological relevance. To rectify this, we sought to understand its biological origin and thus, demonstrate its significance. Our new experimental data conclusively show that lysosomal activity generates the 66–140 fragment from degradation of *soluble* and not fibrillar α -synuclein. This establishes the relevance in studying this Δ N-variant *in vitro*. Briefly, using lysosomes isolated from mouse brains, intact mass spectrometry was used to peptide map products from lysosomal degradation of α -synuclein monomer and fibrils. The results are presented in a new **Figure 1**. We have identified not just 66–140, but also other PD-related truncations (*e.g.* 5–140, 39–140, 65–140, 68–140, Ac1–103, and Ac1–122) being derived from the soluble state by lysosomal activity. We further showed using purified recombinant lysosomal proteases (namely, cathepsin B, L, D and asparagine endopeptidase, AEP), that AEP is the prime candidate in cleaving soluble α -syn at position N65/V66 to generate 66–140 as this protease has a preference to cut at Asn residues (**Figs. S1** and **S2**). We believe this assignment is pertinent because AEP levels/activity is upregulated in PD. Notwithstanding, another lysosomal protease, cathepsin B is also implicated in PD as a genetic risk factor; thus, reaffirms the role of lysosome in synucleinopathies. Finally, we assessed the impact of mouse age on lysosomal protease activity. The rationale is to ask whether α -synuclein truncation generation by lysosomes would be affected as α -synuclein levels are known to increase with age. Brain lysosomes were isolated from mice euthanized at 2 vs. 17 months of age, in which elevation of protease activities from 17 months old brain lysosomes was observed, indicating that the potential for truncations is enhanced with age. This provides additional support with regards to pathological relevance. These figures are reproduced on pages 2–4 of the response letter.

Further, the authors note that there are many charged residues in alpha-syn, but this construct removes the majority of Lys residues, greatly reducing the number of potential salt bridges that could form. It becomes apparent after reading the full paper that this might be the intentional choice for the design of the experiments, but this needs to be more clearly explained in the introduction.

REPLY: We thank the Reviewer for mentioning this important point. While our initial objective was to use this construct to reduce the number of possible charges to interrogate, the newly added data showing 66–140 originates from the proteolysis of soluble α -syn by lysosomal AEP has resulted in a rewrite to the introduction in explaining the rationale of using 66–140. However, we still clearly state the added advantages of this construct with regards in the reduction of charges as well as the homogeneity of fibril population of 66–140.

Although it is true that "nearly all WT alpha-syn fibrils contain at least one salt bridge", most of these involve Lys residues within the region 1-65 among the KTKEGV imperfect repeats.

REPLY: While the Reviewer is correct in pointing out that most of the Lys residues are in the region of 1–65, it should be noted that many of these identified salt bridges involving Lys from this region come from a couple of studies. To clarify the matter, we now only cited papers where charged residues participate in salt bridges are found within the shared α -syn fibril core (residues 36–99). This is updated in the schematic in **Figure 1A**. Please see page 2 of the response letter.

So overall, the choice of construct certainly reduces the significance and generality of the results for disease relevance in this reviewer's opinion. Nevertheless, the study could still be interesting from the perspective of protein biophysics.

REPLY: We hope by adding new insights into the specific protease responsible for generating 66–140 from the soluble state of full-length α -syn has strengthen the relevance of this construct. The broader implication of our lysosomal study is clear as we identified 8 out of the 13 PD-related truncations from patients. Moreover, since we now also show the potential of truncation generation increases with age using a mouse model, there is also an increase disease connection. To the best of our knowledge, the mass spectrometry data from lysosomes and isolated lysosomal proteases presented herein is all new.

2. The data overall are high quality but there are many confounding observations, so the hypothesis(es) are not clearly stated and the experimental measurements, although voluminous, do not seem to be focused on testing hypotheses in a way that arrives at satisfying conclusions. For example from the discussion and concusions there are several statements along these lines

"While it was expected that Ala-substitutions at K80 and E83 would have strong influences as they are found within the ssNMR structure core, the dramatic effects of K96A and K97A were not as they reside outside in the disordered region."

"Inefficiency of fibril propagation of the native 66–140 structure by E83A makes sense because E83A forms a distinct fibril polymorph which is incompatible with 66–140. For K80A, the results are conflicting."

"For K97A, its behavior is puzzling because the data would suggest that it adopts a very similar structure, and its able to cross-seed soluble 66–140"

REPLY: We are pleased that the Reviewer complemented us on the high quality data, and we apologize for any confusion in our writing. In addressing these statements, we have revised large

portions of the Discussion and Conclusion to improve clarity. In addition, we have revised the Results to remove any discussion points, as recommended by Reviewer 1.

Overall, it seems that there are many cases where the seeding, cross-seeding, and structural features do not follow a consistent pattern in the data. Or if there is a pattern, it is indirect.

REPLY: We have revised the manuscript thoroughly and clarified or removed any sentences that may have caused confusion. This significant revision should now make our interpretations clear.

3. Another problematic issue is the discussion of salt bridge measurements comparing the NMR and Raman results. Initially it is noted that Raman bands are more ordered in 66-140 than Ac1-140, such as sharpened amide III (1226 cm^{-1}) and narrower amide-II (1560 cm^{-1}) bands. In addition there is a new peak feature at 1012.5 cm^{-1} , which eventually in the discussion is mentioned as a possible salt bridge feature, but when first raised, there is no attribution to a potential structural feature correlated with the spectral fingerprint. It would be helpful at minimum to be explicit at this point of the manuscript and indicate that the 1012.5 cm^{-1} is postulated to be due to a new salt bridge geometry or environment.

But this is hard to follow since later the authors point out that there is no prior evidence of similar peaks in alpha-syn fibril studies, even though many of them are known to have salt bridges.

REPLY: We appreciate the Reviewer's point regarding the unique peak of 1012.5 cm^{-1} of 66–140. We refrained from discussing in detail the interpretation and tentative peak assignment in the Results and elected to mention it in the Discussion. We apologize if this had caused some consternation. While we would like to please the Reviewer by stating in the Results, Reviewer 1 explicitly recommended not include any discussion in the Results. While it is true that our full-length protein does not show the 1012.5 cm^{-1} peak, even though it contains two salt bridges, a shoulder at 1008 cm^{-1} is now can be seen in the full-length acetylated (Ac) AcK97A and AcK80A fibrils (these mutants were made to address Reviewer 4), highlighting the conformational sensitivity of this spectral region is replicated in the full-length protein (**Fig. S28**). The figure is reproduced on page 7 in the response letter. To define the origin of this peak, it likely will require both experimental and theoretical work, which is outside the scope of this work.

Moreover, although there is a REDOR NMR measurement to indicate that E83 is involved in a salt bridge, the potential assignment of K97 is not a resolved residue in the NMR assignments, which end at 95. Then a series of cross-seeding experiments muddy the waters further, for example where substitutions at K97 do not have the expected effects consistent with an essential salt bridge. So then the argument shifts to K97 potentially having an indirect role. But then how can this be a salt bridge in the structure by NMR and explain the 1012.5 cm^{-1} peak?

REPLY: We thank the Reviewer for bringing this issue to our attention. To make a convincing case in proposing an E83–K97 salt bridge in 66–140, we have made significant revisions to the manuscript. This includes new **Figs. 3C** and **S25**, along with additional text. In brief, we first reanalyzed the NMR data and found that although residues K96 and K97 could not be unambiguously assigned in 66–140, it is clear that the observed NMR signals correspond to more than one lysine. This observation is now stated in the Results. Next, a structural model of 66–140 protomer is shown based on the overlapping sections of assigned residues of 66–140 with the ssNMR structure of 1–140 amplified from human Lewy body dementia (LBD) tissue by Reinstra and co-workers (Nat Commun., 2024, 15(1), 2750). This model is presented in **Figure 3C**. Lastly, molecular dynamics simulations have also been conducted on the model presented in **Fig. 3C** to add more clarity on K97 participating in a salt bridge with E83. This is shown as a new **Figure S25** along with its own separate results section. We hope

these new data have added more weight to support the presence of K97–E83 salt bridge in 66-140 fibrils. Figures have been reproduced below for your convenience.

Figure 3. ssNMR of 66–140 fibrils **(A)** 2D ^{13}C - ^{13}C ssNMR spectra of 66–140 fibrils, recorded under conditions where only immobilized, structurally ordered segments are visible. Residue assignments are based on 2D NCACX/NCOCX, and 3D NCACX/NCOCX/CANCOX spectra (**Figs. S7–S12**). Contour levels increase by successive factors of 1.3. **(B)** Secondary chemical shifts $\Delta\delta^{13}\text{C}_\alpha$ (top) and $\Delta\delta^{13}\text{C}_\beta$ (bottom) obtained by subtracting random-coil shifts from the observed chemical shifts. Blue arrows indicate β -sheet secondary structure predicted by TALOS-N. **(C)** Structural model of the 66–140 fibril generated by modifying model 8 in PDB ID: 8FPT⁷ using UCSF Chimera⁷⁸. NMR-assigned residues consist of 66 to 95, with blue arrows depicting the β -sheet secondary structure for residues 69 to 72, 74 to 79, and 87 to 92. Fibril axis as indicated.

Figure S25. 66–140 model structure by molecular dynamic simulations. **(A)** Starting structure (left) and ending structure from a 30-ns MD simulation. **(B)** Radial distribution functions for the distance of K80 (blue), K96 (purple), and K97 (green) to E83. **(C)** A zoomed-in view of possible salt bridges (green, dotted line) between K97 and E83 in the structure depicted in panel A, left (green box). **(D)** Salt bridges occurring over the 15-ns MD simulation trajectory analyzed, where white indicates a salt bridge was detected. Top to bottom is ranked by the number of salt bridges.

In the end, these are all very interesting observations, and there is the foundation for a potentially impactful study, but as it stands, this manuscript has too many uncertainties and open ends to be ready for publication as a high impact communication.

REPLY: Again, we thank the Reviewer for their thoughtful comments. We hope that the inclusion of new data in our revision has adequately addressed their concerns.

Reviewer #4 (Remarks to the Author):

The authors investigated the role of charged residues in the 66-140 segment of α -synuclein on the aggregation pattern. Morphology of fibrillar aggregates assembled by the wild type 66-140 and asset of mutants was characterized by AFM and TEM. In parallel, the structure of aggregates was probed by NMR and Raman spectroscopies. The major goal of the described experiments was to elucidate the polymorphism of fibrils formed by α -synuclein, but the authors fall short in accomplishing the goal as truncated form of the full-length α -synuclein are different from the ones assembled by the truncated protein. Although replacing charged residues with alanine facilitate the aggregation kinetics, the extension of the described approaches to the full-length α -synuclein is needed to make the results biologically significant. Additionally, there are multiple serious weaknesses in the paper that are listed below.

REPLY: We thank the Reviewer for the suggestion of making the results biologically significant by extending these observations on full-length protein. First, we wanted point out that in addition to extending experiments to the full-length protein, we also added new data to establish the pathophysiological relevance of the construct 66–140 by determining its biological origin (**Figs. 1, S1, S2 and Tables S1–3**). Please see pages 2–4 in the response letter. In brief, we have determined that a specific lysosomal protease, asparagine endopeptidase (AEP), is responsible for the generation of 66–140 by using lysosomes isolated from mice brains. We have also improved the significance of this study by providing additional evidence for the existence of a salt bridge in 66–140 through molecular dynamics simulations (**Fig. S25**). Please see page 13.

In addressing the full-length protein, the importance of three C-terminal K80, K96, and K97 mutants were validated in the acetylated (Ac) full-length protein. These data are presented as a new **Figure 8** and supporting **Figures S26–S28**. Please see pages 5–7 for the reproduced figures. These data show that both AcK96A and AcK97A mutants mirror the sluggish behaviors of the corresponding 66–140 mutants. Interestingly, Ala-substitution of K80 had opposite behaviors for full-length and 66–140, where aggregation kinetics is accelerated and slowed, respectively. Collectively, the results support that 66–140 as an useful model for investigating α -syn fibril formation, offering relevant sites that modulate aggregation propensity.

1. Structure of aggregates and its relation to the morphology of fibrils. Structurally the aggregates were characterized by NMR and Raman spectroscopies and these studies revealed the formation of characteristic structural features of the aggregates such as β -sheets assigned to specific location in the wild type 66-140 protein. However, the link of these structural features with morphologies of fibrils revealed by TEM and AFM imaging is questionable. The yield of fibrils was considerably less than 100%, so the comparison of imaging and spectroscopic data could be made if only fibrils were used for the secondary structure analysis. This concern is critical when similar studies were made for mutants, which were characterized the low yield of fibrils. For example, the yield of fibrils was as low as 13% for K97A mutant.

REPLY: We apologize for the confusion. We need to clarify how these experiments were conducted. For all structural analysis, fibril-containing samples, irrespective of fibril amounts generated post-aggregation were spun down at 100,000 g for 30 min in order to pellet insoluble material. This pelleted material which contained fibrils was used for all structural analysis. The soluble material was discarded and not used in any of these experiments. So, despite low yields of certain mutants, only fibrils in the pelleted fractions were measured and compared.

2. Cross-seeding experiment were used to reveal the differences in structures of fibrils assembled by mutants and the wild-type 66-140 sample. To eliminate the contribution of aggregates formed from the wild-type 66-140 sample in the aggregation process, the authors used the approach at which aggregation process of the wild-type 66-140 sample is fully blocked. The approach is neither

described nor explained, so one could assume that the difference in cross-seeding experiments is due to unexplained features of the aggregation kinetics rather than the blocking the seeding reaction of the wild-type 66-140 aggregation by non-homologous structure of fibrils seeds of the mutant (E83A). The approach should be described and its potential contribution of the aggregation process should be carefully discussed.

REPLY: Again, we apologize for not clearly stating how these experiments were conducted. Since aggregation is facilitated by adding glass beads, we elected to not use beads for seeding experiments. As a result, the unseeded sample does not aggregate and stays soluble (no ThT response) under the time frame of these aggregation experiments. Therefore, when seeds are added, any ThT response is the result of seeding and not *de novo* aggregation. This is now clearly stated in both the Results and Methods and in the Figure Legends.

3. Morphologies of fibrils was studied by AFM and TEM. The selected images are of the high quality, but no analysis of the variability between such feature of twisted fibrils as their periodicity is provided. The periodicity numbers are indicated, but no explanation on the statistics of these measurements is provided. Unbiased analysis is needed to support the numbers on the fibril's periodicity listed in the paper. What is the partition between different morphologies for each protein? What is the variability of the periodicity along the selected fiber, so the morphological heterogeneity was observed for individual fiber? Additional unanswered questions: Are the individual fibrils have only one periodicity or there are segments within each fibril with two different pitch values? How does the periodicity including the partition between the various types vary for different mutants? How does this feature translate into the seeded fibrils? How close are the data obtained with TEM and AFM approaches?

REPLY: We thank the Reviewer for their insights and suggestions regarding the AFM data. First of all, we wanted to note that the various helical pitches reported for each mutant are from analyzing between 11 to 23 fibrils by AFM. Due to the low fibril numbers reported from AFM, we did not at the time provide statistics reporting how many fibrils represent a particular polymorph. This data is now provided for each reported helical pitch. To clarify how helical pitches were calculated, we selected a fibril and averaged all the periodicities along the fibril axis. This is reported along with the SD. From these data, we did not observe significant changes in variability of the periodicity along a selected fibril. This same observation was noted from analyzing TEM images taken from our seeding data.

While the low number of fibrils analyzed by AFM could not conclusively determine polymorph distribution within a given sample, we turned to TEM, to which we could analyze many more fibrils. We now report from analyzing ($n > 50$ fibrils) that similar helical pitches are observed to those reported from AFM. Specifically, these data revealed the 73 ± 5 nm for K80A, 84 ± 3 nm for K96A, 86 ± 3 nm for K97A, 88 ± 2 nm for D98A and 89 ± 3 nm for K102A are the main polymorph. This is reported in **Figure S15**.

4. The link of mutations with the sites of potential salt bridges in aggregates formed by the wild-type 66-140 is discussed but experiments at different ionic strengths could provide the data enabling the authors to test the major hypothesis on the role of the salt bridges in the aggregation process.

REPLY: We thank the Reviewer for this suggestion. We now include a new SI **Figure S3** that reports on aggregation of 66–140 in the absence and increasing NaCl (0 to 0.8 M). The results show that an increase in solution ionic strength results in faster aggregation; both lag and growth phases shortened with increasing NaCl concentration. Corresponding TEM images confirmed that fibrils were made. The figure is reproduced below for your convenience.

Figure S3. Aggregation kinetics monitored by ThT (10 μ M) fluorescence of soluble 66–140 (40 μ M) aggregated in pH 7.4 buffer (20 mM NaPi) containing either 0 (gray), 0.4 (purple), 0.8 (blue) or 1 M (magenta) NaCl at 37 $^{\circ}$ C with continuous linear shaking supplemented with a 2-mm glass bead. Solid lines and shaded regions represent mean and SD, respectively ($n \geq 6$). Representative TEM images of 66–140 fibrils taken post aggregation in 0.4 and 0.8 M NaCl. Scale bar are as shown. Here, the protein was buffer exchanged into milli-Q water, filtered through YM100 membranes, and the appropriate 10 \times buffer for each condition was added.